



# Influence of radiosonde observations on the sharpness and altitude of the midlatitude tropopause in the ECMWF IFS

Konstantin Krüger[1], Andreas Schäfler[1], Martin Weissmann[2], and George C. Craig[3]

[1]Deutsches Zentrum für Luft– und Raumfahrt (DLR), Institut für Physik der Atmosphäre, Oberpfaffenhofen, Germany
[2]Institut für Meteorologie und Geophysik, Universität Wien, Vienna, Austria
[3]Meteorologisches Institut München, Ludwig–Maximilians–Universität, Munich, Germany

*Correspondence to*: Konstantin Krueger (konstantin.krueger@dlr.de)

## Abstract

Initial conditions of current numerical weather prediction systems insufficiently represent the sharp vertical gradients across
the midlatitude tropopause. Data assimilation may provide a means to improve the tropopause structure by correcting the erroneous background forecast towards the observations. In this paper, the influence of assimilating radiosonde observations on the tropopause structure, i.e., the sharpness and altitude, is investigated in the ECMWF IFS. We evaluate 9729 midlatitude radiosondes launched during one month in autumn 2016. About 500 of these radiosondes, launched on request during the North Atlantic Waveguide Downstream impact Experiment (NAWDEX) field campaign, were used to set up an observing
system experiment (OSE). The OSE comprises two cycled assimilation forecast experiments, one with and one without the non-operational soundings. The influence on the tropopause is assessed in a statistical, tropopause–relative evaluation of observation departures of temperature, static stability ($N^2$), wind speed and wind shear from the background forecast and the analysis. The background temperature is overestimated at the tropopause (warm bias, ~1 K) and underestimated in the lower stratosphere (cold bias, –0.3 K) leading to an underestimation of the abrupt vertical increase of $N^2$ at the tropopause. We show
that the increments (differences of analysis and background) reduce background biases and improve the tropopause sharpness. Profiles with sharper tropopause exhibit stronger background biases and, in turn, an increased positive influence of the observations on temperature and $N^2$ in the analysis. Wind speed is underestimated in the background, especially in the upper troposphere (~1 m s$^{-1}$), but the assimilation improves the wind profile. For the strongest winds the background bias is roughly halved. The positive influence on the analysis wind distribution is associated with an increase of vertical wind speed shear, which is underestimated above the tropopause. In addition to the tropopause sharpening, we detect a shift of the analysis
tropopause altitude towards the observations. The comparison of the OSE runs highlights that the main contribution to the tropopause sharpening can be attributed to the radiosondes. This study shows that data assimilation improves wind and temperature gradients across the tropopause, but the sharpening is small compared to the model biases. Hence, the analysis still systematically underestimates the tropopause sharpness which may negatively impact weather and climate forecasts.



# 1 Introduction

The extratropical tropopause is the physical boundary that separates the well–mixed upper troposphere (UT) from the stably stratified lower stratosphere (LS) (e.g., Gettelman et al., 2011). The transition from the UT to the LS is characterized by sharp vertical gradients of temperature, humidity and wind and the strength of these gradients determines the sharpness and altitude of the tropopause. In the UT, the average temperature decreases with altitude towards a minimum at the tropopause. Above the tropopause, a ~2 km thick temperature inversion is followed by a nearly isothermal temperature in the LS. This temperature distribution leads to a rapid increase of the squared static stability ($N^2$) from low values ($1 \times 10^{-4}$ s$^{-2}$) in the UT to high values ($4 \times 10^{-4}$ s$^{-2}$) in the lowermost 2–3 km of the LS referred to as the tropopause inversion layer (TIL; Birner et al., 2002). The $N^2$ maximum above the tropopause (within the TIL) is used as a metric for the tropopause sharpness (e.g., Haualand and Spengler 2021; Boljka and Birner, 2022). The TIL acts as a barrier for vertical transport leading to sharp distributions of trace species, across the tropopause, e.g., of specific humidity (Krüger et al., 2022). The vertical distribution of wind in the midlatitude UTLS is highly variable, but on average wind speed linearly increases with altitude in the troposphere towards a maximum just below the tropopause (e.g., Birner et al., 2002; Birner, 2006; Schäfler et al., 2020). Above, wind speed rapidly decreases in the LS associated with increased vertical shear of the horizontal wind (Birner, 2006; Schäfler et al., 2020).

Temperature and wind gradients directly determine the potential vorticity (PV) distribution, which is additionally influenced by humidity–driven radiative modification of the temperature gradients (e.g., Ferreira et al., 2016). The strong meridional PV gradient near the tropopause acts as a waveguide for Rossby–waves (Schwierz et al., 2004; Martius et al., 2010) and, in turn, impacts downstream weather development in the midlatitudes (Harvey et al., 2018). Thus, an accurate representation of the sharp cross–tropopause gradients in the initial conditions is of high importance for numerical weather prediction (NWP) models. However, forecast PV gradients rapidly decline within short (12–24 h) lead times (Gray et al., 2014; Lavers et al., 2023) which is attributed to a smoothing effect of the advection scheme that dominates sharpening effects of parameterized processes; such as radiative cooling driven by water vapor, microphysics, and turbulent mixing (Saffin et al., 2017). The weakening PV gradients are likely associated with background forecast errors of temperature, humidity and wind at the tropopause, which may affect the quality of the analysis. At the tropopause, Bland et al. (2021) found a warm bias (few tenths of K) in analyses of the European Centre for Medium–Range Weather Forecast's (ECMWF) Integrated Forecasting System (IFS). The presence of a warm bias in IFS short–range forecasts and analyses near the tropopause was also indicated in earlier studies (Bonavita, 2014; Ingleby et al., 2016). In the LS, a moist bias (e.g., Krüger et al., 2022) leads to a cold bias at altitudes between 0.5 and 2 km above the tropopause in the IFS (Bland et al., 2021). Schäfler et al. (2020) indicated a systematic underestimation of jet stream wind maxima and showed large wind errors of up to 10 m s$^{-1}$ for individual cases in IFS short–range forecasts and analyses. Lavers et al. (2023) detected a vertically increasing slow wind bias in IFS background in the troposphere (up to roughly 0.6 m s$^{-1}$). This wind speed underestimation in models is in line with Birner et al. (2002) who found notably underestimated UT wind maxima and vertical wind shear in the ERA–15 reanalysis. Quantitative assessments of the



magnitude of vertical wind shear at the tropopause revealed an underestimation by a factor of 2–5 (Houchi et al., 2010; Schäfler et al., 2020).

Data assimilation (DA) has shown a positive influence on the analysis in the UTLS, i.e., a reduction of the short–range forecast
errors of temperature (e.g., Radnoti et al., 2010; Bonavita, 2014) and wind (e.g., Weissman and Cardinali, 2007; Weissmann et al., 2012; Lavers et al., 2023; Martin et al., 2023). Two dedicated studies elaborated the influence of DA on the tropopause sharpness. Birner et al. (2006) investigated the role of satellite DA in the Canadian Middle Atmosphere Model (CMAM) providing a vertical resolution of 1 km near the tropopause and used a 3DVAR assimilation scheme. A decrease of $N^2_{max}$ in an experiment with assimilated satellite observations compared to a model run without suggested that DA smears out the gradients
near the tropopause. A more recent study by Pilch Kedziersky et al. (2016) analysed the influence on the tropopause sharpness at the positions of GPS radio occultation (GPS–RO) observations in ECMWF's ERA–Interim reanalysis and IFS analysis, using 4DVAR (e.g., Rabier et al., 2000) and a vertical resolution of ~500 m at the tropopause. The detected increase of $N^2$ in an ~1 km thick layer just above the tropopause and a decrease of $N^2$ above and below this layer is corresponding to a tropopause sharpening, which was attributed to the assimilation of GPS–RO data. Both studies differ in terms of the applied methods to
diagnose the influence, the used observation type, the spatial resolution and the DA schemes. Hence, no definitive conclusion can be drawn as to whether DA sharpens or smoothens the tropopause. Additionally, it should be noted that both studies are based on variational DA schemes without a flow-dependent estimate of the error covariance matrix (**B**) which balances the background and the observations by accounting for their estimated errors (e.g., Bannister et al., 2008). Flow-dependent estimates of B as they are nowadays used in the hybrid DA scheme of ECMWF are expected to lead to more accurate increment
structures and therefore a better representation of sharp gradients.

In this study, we investigate radiosonde profiles, which provide highly resolved and accurate profiles of temperature and wind components (e.g., Vaisala, 2017), and thus are suitable to resolve the sharp vertical gradients at the tropopause. The measured quantities are directly assimilated and, although they only account for a small proportion (about 2 %) of the total assimilated meteorological information, they contribute to a 5 % reduction in 24-h forecast error in the ECMWF IFS in a statistical sense
(Pauley and Ingleby, 2022). In addition, radiosondes serve as anchor observations for the variational bias correction e.g., for satellite observations, highlighting their important role for DA (Cucurull and Anthes, 2014). The impact of individual observation capabilities such as radiosondes is typically assessed by performing observing system experiments (OSEs; e.g., Bonavita, 2014), e.g., during special observation periods related to field campaigns (e.g., Weissmann et al., 2012; Schindler et al., 2020; Borne et al., 2023). The DA impact can be studied either in model space by using 3D gridded model output or in
observation space, which is the 4DVAR model output (observations and departures) representative for the position and time of the assimilated observation. The latter method has the advantage that a comparison of the observations and departures allows the influence of individual measurement types and parameters in the NWP system to be evaluated.

The scope of this study is to quantify the change of the tropopause structure from the first–guess to the analysis and to relate it to the assimilation of radiosondes. For this purpose, we make use of the 1–month campaign period of the North Atlantic
Waveguide Downstream impact EXperiment (NAWDEX; Schäfler et al., 2018) in autumn 2016 during which 9729 radiosonde





profiles in a region between eastern North America and Europe were assimilated. 497 of the 9729 radiosondes were non–operationally launched and applied in an OSE, which consists of two cycled IFS runs, one with and one without the additional observations (Schindler et al., 2020). The statistical evaluation is performed in a tropopause–relative framework which is mandatory to preserve the outlined sharp gradients in the UTLS when averaging profiles with different tropopause altitudes
(Birner, 2006). As no humidity data at and above the tropopause is assimilated (Bland et al., 2021), we restrict the analysis to profile observations of temperature and wind. We address the following specific research questions:

1. How is tropopause sharpness represented in background forecasts and what is the influence of DA on the analysis? Does the diagnosed temperature and wind influence depend on the tropopause structure and vary in different dynamic situations?
2. Does the influence on the temperature profile affect the tropopause altitude?
3. Can the diagnosed influence be attributed to the assimilated radiosondes or do other observations also affect the tropopause structure?

## 2 Data and Methods

### 2.1 Description of the data set and the OSE

In this study, we analyse about 9200 radiosonde profiles (Fig. 1a) that were routinely measured at 581 sites covering a wide area between North America and Europe from the subtropics to high latitudes (30°N–85°N, 95°W–30°E) during a one–month autumn period (17 September–18 October 2016). The majority of these observations (96 %) were performed at 200 land–based stations while a minor share (4 %) are ship–based observations at 381 variable positions across the North Atlantic. In addition to the routine profile observations, about 500 extra radiosondes were launched in the course of the NAWDEX field campaign
(Fig. 1b), which had the aim to better explore the influence of diabatic processes on the polar jet and weather downstream (Schäfler et al., 2018). Over Europe the extra, on demand radiosondes were released in a variety of synoptic situations, for instance, in diabatically–active warm conveyor belt flows associated with cyclones or in upper–level ridges associated with blocking situations. Six stations over Canada, upstream of the NAWDEX operation region, released two additional radiosondes per day. In addition to the radiosonde observations, more than 700 dropsondes were released from research aircrafts during
the NAWDEX period (mostly in the subtropical and tropical west Atlantic, see Schindler et al., 2020). Due to the low data coverage of the dropsondes above and at the tropopause related to the limited flight altitude of the aircrafts, we restrict our analysis to the radiosonde profiles. All radiosonde and dropsonde profiles were made available for operational assimilation at weather centres (Schäfler et al., 2018). Figure 1 shows the launching position of those radiosondes that were assimilated within the IFS.



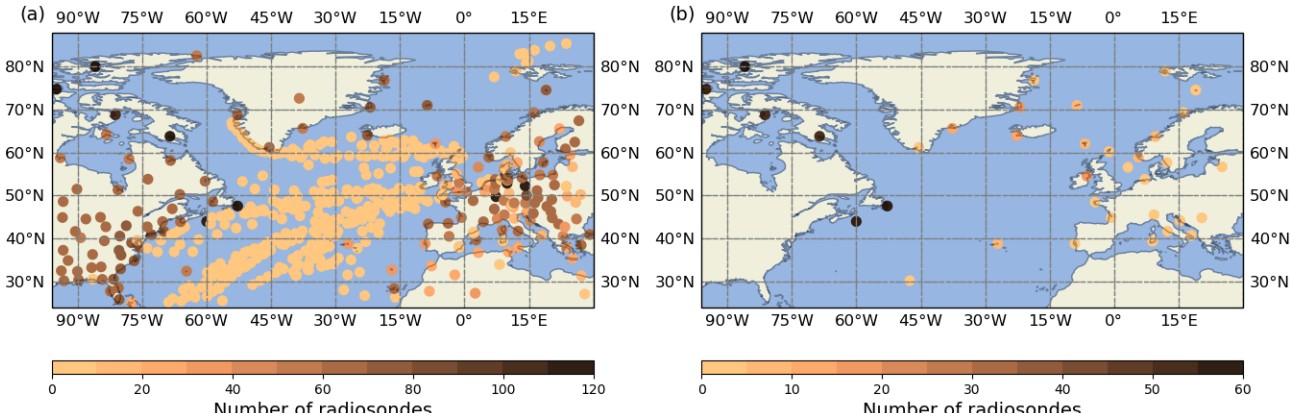

**Figure 1:** Positions of radiosonde launches that were assimilated by the ECMWF IFS between 17 September and 18 October 2016 for (a) all radiosondes (9729), and (b) the subset of 497 non-operational radiosondes launched during NAWDEX. The colouring denotes the number of assimilated profiles at a particular site. The scale of the colourbar changes between (a) and (b).

With the aim to investigate the influence of the extra radiosonde observations during NAWDEX a dedicated OSE was performed with the IFS (Schindler et al. 2020). The cycled OSE covers the whole NAWDEX campaign period (17 September to 18 October) and uses IFS model cycle 43r1 (Cy43r1; ECMWF, 2016), which became operational in November 2016. The triangular–cubic–octahedral grid ($T_{Co}1279$) provides a horizontal resolution of ~9 km and 137 vertical sigma–hybrid levels that range from the surface up to ~80 km. The vertical resolution is highest in the planetary boundary layer decreases with altitude. At typical midlatitude tropopause altitudes (6–15 km; e.g., Schäfler et al., 2020) the vertical grid spacing is about 300 m. The incremental hybrid 4DVAR DA scheme used at ECMWF assimilates observations available in a 12 h time window to update a prior short range forecast in order to achieve the best possible estimation of the atmospherics' state, which is the analysis. More details about the implementation of 4DVAR in the IFS are given in Rabier et al. (2000) or in the IFS documentation (ECWMF, 2016). As in the operational ECMWF system, the B-matrix for the experiments is based on a blended combination of a climatological estimate and an estimate from an ensemble of data assimilations (EDA). The cycled OSE comprises two separate model runs. The control run (CTR) considered all routine and extra radiosondes as well as the dropsondes launched during NAWDEX. The denial (DEN) run excluded all additional observations in a region over the North Atlantic (25°–90°N; 82°W–30°E). In addition, a 25–member EDA experiment was conducted at lower horizontal resolution ($T_{Co}639$ ~18 km) for both experiments. More details on the OSE design are given in Schindler et al. (2020).

For our analysis we retrieved observation feedback files of the OSE experiment from ECMWF's observation database (ODB), which contain the (radiosonde) temperature and wind observations and their departures from the background and analysis state given as profiles using pressure used as the vertical coordinate. On the one hand we analyse the influence of all 9729 radiosondes in the operational CTR run. On the other side, the influence of the subset of 497 radiosondes in the CRL is compared with the DEN experiment, where they were excluded and only passively monitored. The observation space data is stored during the 4DVAR process at the position and time of the observations. It has to be noted that the radiosonde profiles





are not assimilated at their fully measured vertical resolution (which would be ~5 m) but at a limited number of levels (~50–350), which depends on the reporting type (e.g. alphanumeric, BUFR, highres. BUFR) the individual stations used for the data transmission to the Global Telecommunications System (GTS) (Ingleby et al., 2016). Cy43r1 does not consider the horizontal drift of the radiosondes. From the requested feedback files, we extract those observations that are actively assimilated. Some

profiles (<1 %) which do not provide temperature and wind data above the 540 hPa level (~5 km) are excluded from the statistical analysis. This level is selected as it serves as a starting point for the tropopause detection (Sect. 2.2).

**2.2 Data processing and tropopause-relative coordinates**

First, the observation space background (or first guess, $y_{FG}$) and analysis ($y_{AN}$) states are derived from the observations ($y_O$) and departures from the first–guess ($dep_{FG}$, referred to as innovation) and the analysis ($dep_{AN}$, hereinafter residuals) as follows:

160             Innovation: $dep_{FG} = y_O - y_{FG}$                                         (1),

                Residual:   $dep_{AN} = y_O - y_{AN}$                                          (2).

The observation space increment is defined as the analysis minus the background state and shows of whether a quantity has been increased or decreased in the DA cycle:

                Increment $= y_{AN} - y_{FG}$                                                 (3).

In a next step, we derived the geometrical altitude from the pressure data based on the hydrostatic equations given in ECMWF (2016). The observation and model states are then linearly interpolated in the vertical to an equidistant 10 m grid. The potential temperature ($\theta$) and the squared static stability ($N^2$) are computed from the temperature profile using

$$N^2 = \left(-\frac{g}{\theta}\right) * \left(-\frac{d\theta}{dz}\right) \left[\frac{1}{s^2}\right]$$                (4),

with the vertical gradient of $\theta$ ($\frac{d\theta}{dz}$) in geometrical coordinates (z) and the gravitational acceleration (g; g = 9.81 ms$^{-2}$).

From the wind profiles, the vertical wind shear as

$$\text{Wind shear} = \left(\frac{d|\vec{u}|}{dz}\right) [s^{-1}]$$                                 (5),

with the vertical gradient $\frac{d|\vec{u}|}{dz}$ of the horizontal wind vector $\vec{u}$.

Various tropopause definitions are used in the literature, which are defined based on the particular thermal, dynamic and chemical characteristics of the UTLS (e.g., Gettelman et al., 2011). We rely on the lapse–rate tropopause (LRT), which, by

definition, points to the sharp transition of thermal stratification from the UT to the LS (e.g., Birner et al., 2002; Tinney et al., 2022). The LRT is defined as the lowest level at which the lapse rate (i.e., the vertical temperature gradient) falls below 2 K km$^{-1}$, subject to the condition that the average lapse rate from that level to any point within the overlying 2 km layer does not exceed 2 K km$^{-1}$ (World Meteorological Organisation (WMO), 1957). The WMO definition also comprises a further criterion to determine a secondary (or "double") tropopause, however, in this analysis we only determine the "first" LRT. The LRT

altitude is used to determine LRT–relative altitudes ($z_{LRT-relative}$) for each radiosonde profile, which is the difference of the geometrical height profile ($z_{geometrical}$) and the LRT altitude ($z_{LRT}$) following Eq. 6:



$$z_{LRT-relative} = z_{geometrical} - z_{LRT} \tag{6}.$$

In the statistical assessment averages of parameters and increments are calculated in tropopause–relative coordinates (e.g.,

Birner et al., 2002). Although the LRT definition permits a robust detection of the tropopause altitude for most atmospheric

conditions, the 2–K criterion entails some important limitations (for details see Tinney et al., 2022 and references therein). First, the 2–K threshold can lead to undesired false detections of the LRT altitude (in the following referred to as "misdetection") at small temperature fluctuations which are often present in the lower troposphere (boundary layer) but also occur in the mid-troposphere. To avoid LRT misdetections, the tropopause detection is performed above ~5 km (540 hPa) altitude. Second, in situations of weak vertical temperature gradients, i.e. smooth transitions across the tropopause, the 2–K

threshold is sometimes not quite met leading to LRT jumps by several kilometres for neighbouring, similar temperature profiles (Krüger et al., 2022). This occurs typically in the vicinity of the jet streams where the tropopause altitude shows a discontinuity or double tropopauses may occur (Hoffmann and Spang, 2022; Tinney et al., 2022). Hence, a slightly different temperature representation in models and observations can result in large LRT altitude differences. The potential influence of the misdetections is discussed in Sect. 3.2.2 and 4.

LRT altitudes are derived individually for the observations, (in the following: $LRT_{yO}$) the background ($LRT_{yFG}$) and analysis ($LRT_{yAN}$) by following the WMO definition outlined above. Figure 2 illustrates the vertical distribution of $LRT_{yO}$ for 9729 profiles which has a bi–modal shape in the altitude range from 6 km to 18 km with peaks at 11.5 and 15.5 km. The highest frequency, which occurs for the lower maximum, represents midlatitude profiles at 10–13 km altitude (see colouring in Fig. 2), which are typical LRT altitudes expected from climatology in the midlatitudes in autumn (e.g., Hoffmann and Spang, 2022;

Krüger et al., 2022). The broad spectrum of the tropopause altitudes below 15 km is related to the variability of the midlatitude tropopause in different of synoptic situations, e.g., in ridges and troughs. A second, smaller maximum of $LRT_{yO}$ at 15–16 km is related to profiles at low latitudes in the vicinity of the subtropical jet.

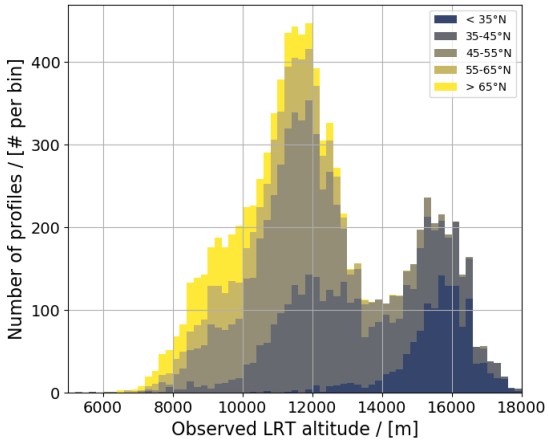

**Figure 2:** Stacked distribution of $LRT_{yo}$ with 0.2 km bin size for all 9729 radiosondes. The colouring shows the latitude of the radiosonde

stations (10° bins).



Figure 3 presents the mean vertical profiles of observed temperature, $N^2$, wind speed and wind shear profiles averaged in $LRT_{yo}$–relative (with respect to the observed tropopause) coordinates. These profiles outline the main characteristics of the midlatitude tropopause that are known from climatology (e.g., Birner et al., 2002; Grise et al., 2010; Hoffmann and Spang, 2022): Above a linearly decreasing temperature in the troposphere (~7 K km$^{-1}$), a temperature minimum of about 213 K is reached at the $LRT_{yo}$. Above, a distinct temperature inversion (0–1.5 km above $LRT_{yo}$) follows before the temperature becomes roughly isothermal (up to ~5 km above the LRT) in the stratosphere. This change of stratification results in a rapid jump of $N^2$ (from 2 to 6.5x10$^{-4}$ s$^{-2}$) across the $LRT_{yO}$ altitude. Wind speed continuously increases with altitude in the troposphere up to a maximum (~23.5 m s$^{-1}$) at ~1 km below $LRT_{yO}$. Corresponding to the distribution of wind speed, the vertical wind speed shear (in the following referred to as wind shear) is positive up to the wind speed maximum, then abruptly decreases beyond and reaches a distinct minimum (~5x10$^{-3}$ s$^{-1}$) at about 300 m above $LRT_{yO}$. Please note that the presented data set of 9729 radiosondes provides a high data coverage (Fig. 3a, blue line) in the UTLS.

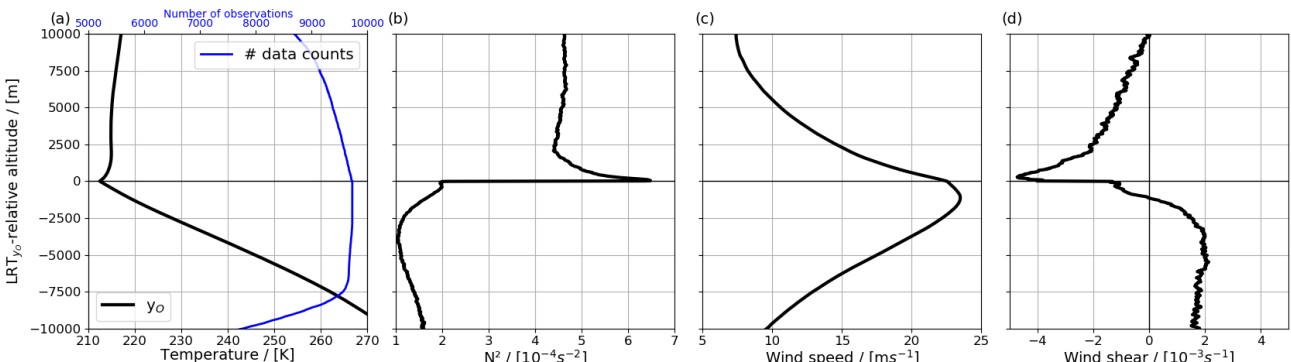

**Figure 3:** $LRT_{yO}$–relative mean profiles of (a) temperature (black) and number of data (blue), (b) $N^2$ and (c) wind speed and (d) wind shear using 9729 radiosondes.

## 3 Results

### 3.1 Increments in geometrical and tropopause–relative coordinates

Figure 4 shows the time series of temperature increments over Iqaluit, Canada between 17 September and 18 October 2016, in both geometrical (Fig. 4a) and $LRT_{yO}$–relative coordinates (Fig. 4b). Iqaluit is selected as it comprises a high number of radiosonde profiles (#114, at a 6–hourly interval) and outlines the typical high tropopause altitude and wind speed variability related to the changing synoptic situations. Several strong jet stream events with wind speeds of occasionally >45 m s$^{-1}$ passed over the station, which are accompanied by high variability of the LRT (7–13 km).



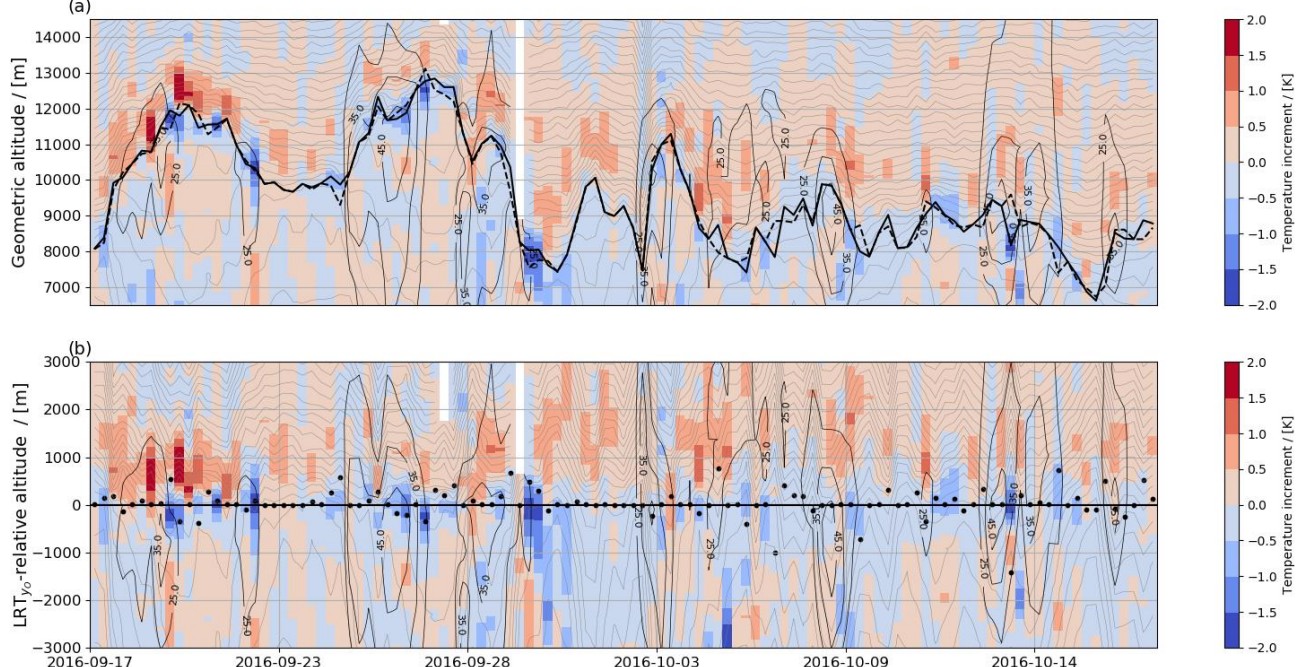

**Figure 4:** Time series (17 Sep. –18. Oct. 2016) of temperature increments (colour shading) at Iqaluit (63.75°N, 68.53°W, Canada) illustrated in (a) geometrical height and (b) $LRT_{yO}$–relative coordinates. The panels are superimposed by the observed θ (thin grey lines, $\Delta\theta = 4K$) and wind speed (thin black contours). In (a) the black thick (dashed) line show $LRT_{yO}$ ($LRT_{yFG}$) and the black dots in (b) show their difference.

In geometrical altitude strong positive (>1 K) and negative temperature increments (< –1 K) are stacked and roughly follow the tropopause. Due to the variable LRT altitude, averaging of the profiles in geometrical coordinates would blur the vertical distribution of the increments and thus, hide a potential influence on the tropopause in a statistical evaluation. However, in $LRT_{yO}$–relative coordinates, the negative increments can be clearly assigned to the about ±0.5 km around the tropopause, and the positive increments to 2 km thick layer above. The vertical extent of the positive and negative increments is relatively persistent along the entire time series, but the magnitude is variable. The distance between observed and background tropopause altitude is mostly in the range of ~100 m, but there are also cases with large altitude differences (>1 km, see discussion in Sec. 2.3).

## 3.2 Statistical assessment

### 3.2.1 Mean tropopause-relative influence

Figure 5 presents $LRT_{yO}$–relative average profiles of temperature, N², wind speed and vertical wind shear for the 9729 radiosondes and their model equivalents. The minimum temperature detected in a layer of ~ ±500 m around $LRT_{yO}$ is overestimated (up to 1 K) in the background profiles (Fig. 5a) confirming a warm temperature bias at the tropopause. In the



LS above, the background temperature decreases less strongly which results in a cold model bias between 0.5–2 km above tropopause. The weaker thermal gradients in the background are accompanied by an underrepresentation of the amplitude and sharpness of the N² jump across the tropopause (Fig. 5b). Wind speed is underestimated in the background throughout the UTLS (Fig. 5c), with a maximum underestimation (0.5 m s$^{-1}$) between –1 km and 0.5 km. The magnitude of wind shear,

respectively its rapid decrease at the tropopause is much weaker in the background, leading to an underestimation of wind shear below and an overestimation above LRT$_{yO}$. Figures 5 a–d show that the analysis is drawn towards the observations for all parameters at any altitude of the UTLS. The slightly sharper tropopause structure reveals a positive influence of DA on the representation of the tropopause in the analysis.

Figures 5e–h shows the vertical structure of the increments. The temperature increments (Fig. 5d) imply a cooling (up to –0.25

K), between –1 km and +0.5 km around the LRT$_{yO}$, i.e., the altitude range of the warm bias. In the LS, a warming of up to 0.25 K between 0.5 and 2 km above LRT$_{yO}$ counteracts the cold bias in the model background. This impact on the temperature distribution results in negative N² increments (–0.15x10$^{-4}$ s$^{-2}$) in a 1.5 km thick layer below the LRT$_{yO}$ and between 1 and 2 km above LRT$_{yO}$ (Fig. 5e). In the 1 km layer above LRT$_{yO}$, N² increments are positive with a distinct maximum of ~0.32x10$^{-4}$ s$^{-2}$ at ~0.5 km. Wind increments are predominantly positive in the entire UTLS (Fig. 5g), which indicates a wind speed

increase in the analysis. The wind increments are stronger in the UT than in the LS peaking (0.25 m s$^{-1}$) at the altitude of the wind maximum and the strongest underestimation in the background (1 km layer below LRT$_{yO}$, Fig. 5c). Increments of wind shear are positive in the 2 km below, and negative in the 1 km layer above the LRT$_{yo}$ (minimum at ~500 m above LRT$_{yO}$). Between ~1.5 to 3 km above LRT$_{yO}$ are positive of comparable magnitude to the increments in the UT.

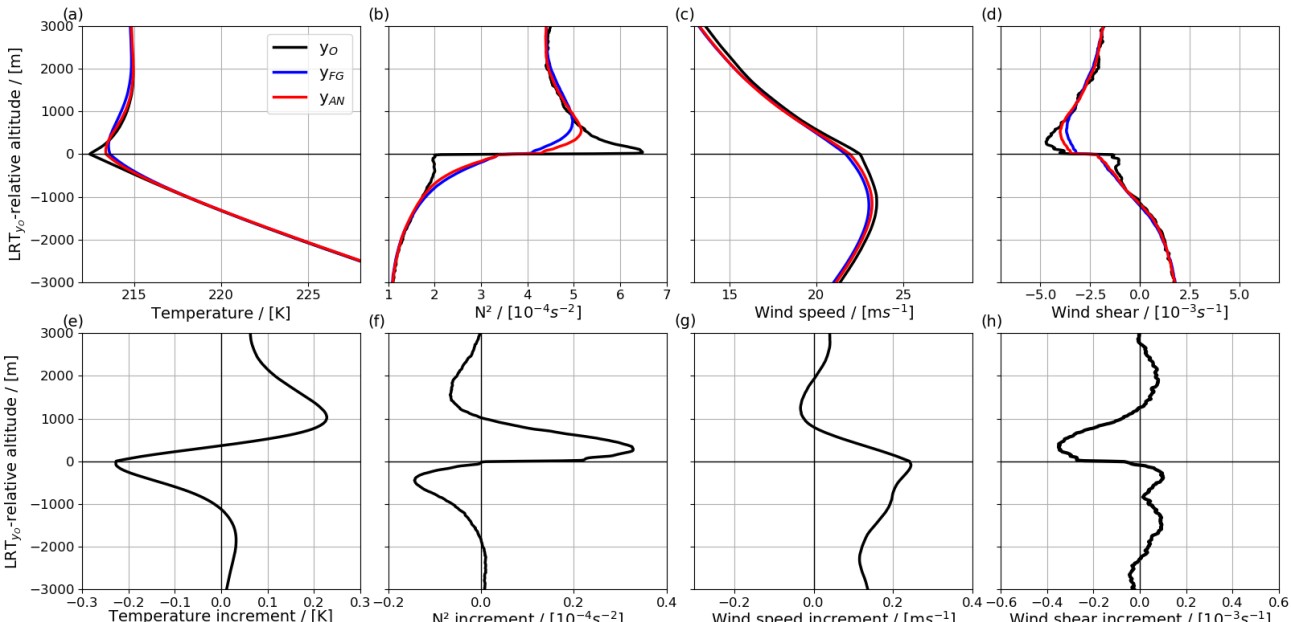

**Figure 5:** LRT$_{yo}$–relative distributions of (a) temperature, (b) N² (c) wind speed and (d) wind shear and the respective increments (e–h; Eq. 3) averaged for the 9729 profiles of observations (black), background (blue) and analysis (red).



### 3.2.2 Sensitivity to the LRT–relative coordinate

Figure 5 presented profiles of the parameters and increments relative to the observed tropopause ($LRT_{yO}$). However, from Fig. 4 we have seen that the observed tropopause altitude may differ from the background (and analysis). This raises the question which LRT–relative view is the most suitable reference for evaluating the tropopause structure. In the following, we present the distributions in different LRT–relative reference systems and discuss their significance to evaluate the tropopause sharpness. Figure 6 illustrates the profiles of temperature and $N^2$ in the observations (Fig. 6a, e), background (Fig. 6b, f) and analysis (Fig. 6c, g) with respect to the observed $LRT_{yO}$, the background $LRT_{yFG}$ and the analysis $LRT_{yAN}$, respectively. In each case the lowest tropopause temperature as well as the strongest temperature inversion and jump in $N^2$ occurs when the "own" LRT is used. This is particularly obvious for the observed profile relative to $LRT_{yO}$ ($y_O(LRT_{yO})$; black curve in Fig. 6a, e). In addition, the background and analysis profiles have the lowest tropopause temperature and strongest inversion when viewed relative to $LRT_{yFG}$ (medium red curve in Fig. 6b) and $LRT_{yAN}$ (light blue curve in Fig. 6c), respectively. Figure 6d and h show the increments referenced to the different LRT–relative coordinates. Each of the LRT reference systems confirms a cooling near the LRT, a warming in the LS above (Fig. 6d), and an increase of static stability just above the LRT (Fig. 6h). The differing LRT altitudes of the individual profiles in $y_O$, $y_{FG}$ and $y_{AN}$ result in small differences in the magnitude of the increments for the different LRT-relative coordinates (discussed in further detail in Sect. 3.3). The increments are smallest when referenced to $LRT_{yFG}$.

As the own–LRT–relative distributions provide highest sharpness, we also consider own–LRT–relative increments (grey line in Fig. 6 d, h) to further analyze the influence on tropopause sharpness. These increments in $LRT_{own}$–relative coordinates, which are calculated as $y_{AN}(LRT_{yAN}) - y_{FG}(LRT_{yFG})$, and ideally remove effects on the average increments from differing $LRT_{yFG}$ and $LRT_{yAN}$ altitudes, have a comparable structure in the LS. However, they show only a slight cooling (<0.1 K) at the tropopause and an increasing warming with decreasing altitude in the troposphere, which does not agree with increments in geometrical and $LRT_{yO}$ space (e.g., Fig. 4 and Fig. 6a). The warming in the troposphere is a systematic temperature bias that is caused by the tropopauses detected at different altitudes (either in $LRT_{yFG}$ or in $LRT_{yAN}$ or both, Fig. 6a). To emphasize the role of differing tropopause altitudes on the distribution of the increments, the four types of increments are shown for cases with similar LRT altitudes (within ±100m), which are almost identical (see overlapping dotted lines in Fig. 6d). We do not further pursue the analysis of $LRT_{own}$–relative increments because such increments are determined after shifting the profiles with respect to the own LRT which does not correspond to real changes to the model background field in geometrical space (when the $LRT_{yFG}$ and $LRT_{yAN}$ differ). We nonetheless present this approach in order to emphasize the sensitivity of cross–tropopause distributions and their increments to the choice of the LRT–relative reference and the impact of systematic LRT altitude differences. As the LRT derived from the radiosondes profiles provides the most realistic representation of the tropopause altitude (Fig. 4a, Fig. 6a), $LRT_{yO}$ is used in the following to analyze the influence on tropopause sharpness (see Sect. 3.2). In addition, the influence on the tropopause altitude is studied relative to the $LRT_{yFG}$ (Sect. 3.3).





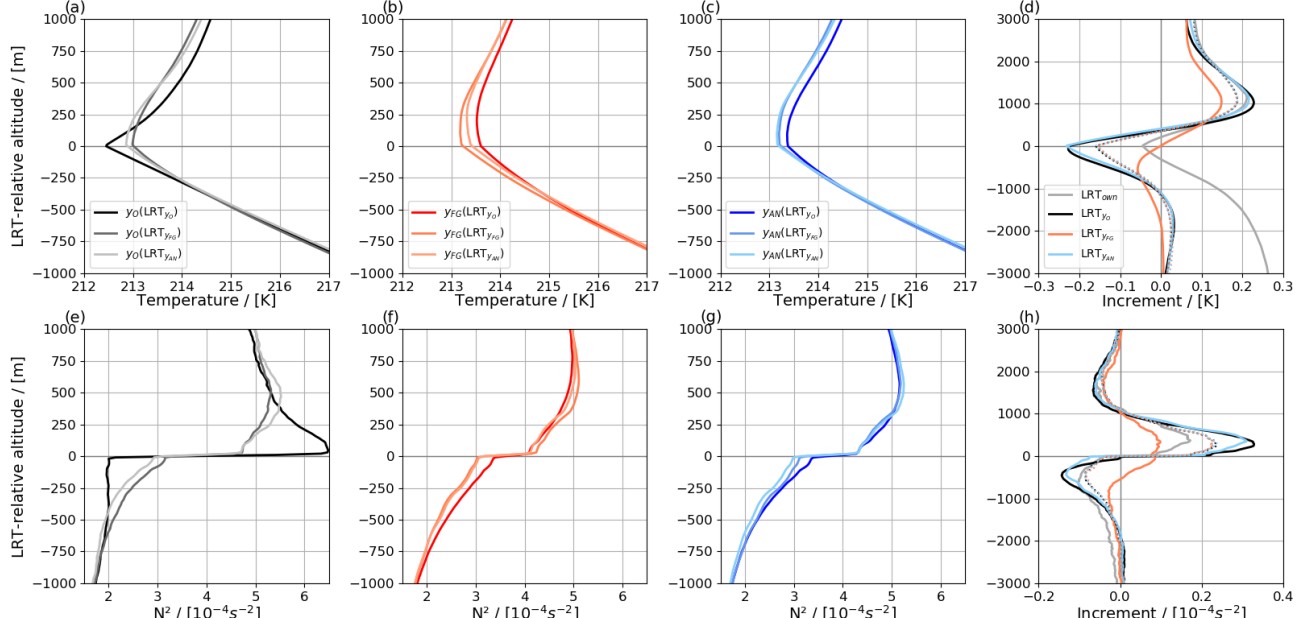

**Figure 6:** Mean profiles of temperature (a–c) and $N^2$ (e–g) for observations (a, e;), background (b, f) and the analysis (c, g) relative to $LRT_{yO}$, $LRT_{yFG}$ and $LRT_{yAN}$, respectively (colour coded). The panels d) and h) show the associated increments. In addition, increments using the own LRTs are shown (grey, calculated as $y_{AN}(LRT_{yAN}) - y_{FG}(LRT_{yFG})$, for details see text). The dotted lines in d) and h) represent the 3712 profiles with the LRT altitudes of observations, background and analysis being within ±100 m (note that dotted lines overlap).

### 3.2.3 Influence on tropopause sharpness

The previous results indicated an increase of sharpness in the analysis and the time series in Fig. 4 suggested high temporal variability of the increments that is likely influenced by particular dynamical situations. Figure 7a illustrates the distribution of the observed maximum squared static stability ($N^2_{max}$) in the 3 km above the $LRT_{yO}$, which is a common indicator for tropopause sharpness (Birner et al., 2006; Pilch Kedziersky et al., 2015). $N^2_{max}$ shows an uni–modal, positively–skewed distribution ranging from 3 to $30 \times 10^{-4}$ $s^{-2}$ with largest frequency (>200 profiles per bin) of between $6–12 \times 10^{-4}$ $s^{-2}$ and lowest frequency (<50 profiles per bin) for $5 \times 10^{-4}$ $s^{-2} < N^2_{max}$ and $N^2_{max} > 15 \times 10^{-4}$ $s^{-2}$. The quartiles of this distribution are used to classify the data into the smoothest ($N^2_{maxQ00–Q25}$), the intermediate ($N^2_{maxQ25–Q75}$) and the sharpest ($N^2_{maxQ75–Q100}$) tropopause cases. The observed profiles (Fig. 7b) display that the sharp class has the lowest tropopause temperature and the strongest inversion with the largest jump of $N^2$. On the contrary, the smoothest tropopauses exhibit a higher tropopause temperature, a weaker temperature inversion and a lower amplitude in $N^2$ at the $LRT_{yO}$. The intermediate class depicts a comparable tropopause structure to the full data set average described in Sect. 3.2.1.



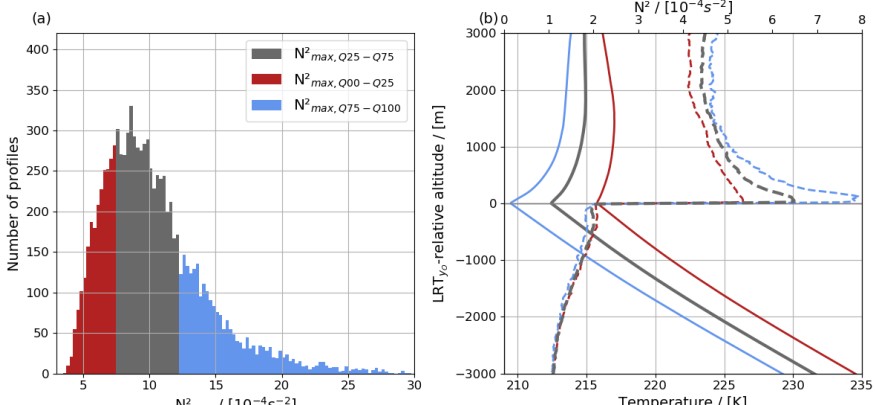

**Figure 7: (a)** Distribution of $N^2_{max}$ as observed in the 3 km layer above $LRT_{yO}$ with bin size 0.5x $10^{-4}$ s$^{-2}$. Quartiles of the $N^2_{max}$ distribution in (a) are used to classify tropopause sharpness: smoothest (red; Q00–Q25), intermediate (grey; Q25–Q75) and strong sharpness (blue; Q75–Q100). **(b)** the corresponding $LRT_{yo}$–relative mean profiles of observed temperature (solid) and $N^2$ (dashed).


For each class of $N^2_{max}$, the mean vertical profile of innovation (Eq. 1), increment (Eq. 3) and residual (Eq. 2) for temperature and $N^2$ relative to $LRT_{yO}$ are presented in Fig. 8. We first focus on the intermediate tropopause sharpness. In the UT, the temperature innovations are weak, negative and vertically nearly constant (Fig. 8a, about –0.1 K) before they reach a minimum of about –1.2 K at $LRT_{yO}$ indicating a warm bias at the background tropopause. Above, the innovations strongly increase and become positive at ~0.5 km above the $LRT_{yO}$ before a maximum cold bias of ~0.3 K is reached at 0.8 km altitude. The temperature increments (Fig. 8b) correspond to the findings in Fig. 5 with the negative increments around the $LRT_{yO}$ counteracting the warm bias, and the positive increments above decreasing the cold bias. In the ±0.5 km around $LRT_{yO}$ large $N^2$ innovations between –2 to 3x10$^{-4}$ s$^{-2}$ illustrate the strong underestimation of tropopause sharpness in the background (Fig. 8d). The average positive (above $LRT_{yO}$) and negative (below $LRT_{yO}$) $N^2$ increments (Fig. 8e) for the intermediate profiles agree in shape and magnitude to the structure of $N^2$ increments given in Fig. 5. Apparently, they lead to a sharpening of the tropopause. Increments are much smaller than the innovations (~20 % for temperature and 10 % for $N^2$) which explains that the vertical structure of the innovation is preserved in the residuals (Fig. 8c, f). For the smooth and sharp classes (blue and red lines in Fig. 8), innovations, increments and residual have a similar vertical distribution but show weaker, respectively stronger amplitudes. For instance, temperature increments are about –0.3 K (–0.1 K) at the tropopause for the sharp (smooth) class and about 0.3 K (0.1 K) for the maximum above, in the LS. The influence is stronger where the background biases are strongest.





**Figure 8:** LRT$_{yO}$–relative mean profiles of innovations, increments, and residuals for (a–c) temperature and (d–f) N² for the classes of N²$_{max}$ defined in Fig. 7.

Figure 9 investigates the variability of wind speed in the radiosonde data set, which varies strongly at the individual stations
over time (Fig. 4). Average wind speeds in a layer of ±3 km around LRT$_{yO}$ range from nearly 0 to 60 m s$^{-1}$ with the highest frequency between 5–25 m s$^{-1}$ (Fig. 9a). Quartiles of mean wind speed divide the data set into weak (wind$_{Q00-Q25}$), intermediate (wind$_{Q25-Q75}$) and strong (wind$_{Q75-Q100}$) winds. Consequently, the weak wind class shows vertically fairly constant low wind speeds (<10 m s$^{-1}$). While the intermediate wind class exhibits a comparable structure to the full data set (Fig. 5), the strongest wind class depicts a pronounced wind maximum (>40 m s$^{-1}$) at –1 km altitude below the LRT$_{yO}$ expressing strong jet stream
winds. The mean wind profiles for each class are shown in Fig. 10. The positive wind innovations of all classes across the UTLS express the underestimated wind speeds in the background (see also Fig. 5). Innovations in the UT are generally larger than above in the LS and peak at the tropopause. Maximum innovations range between 0.5 m s$^{-1}$ for the weak and 1.2 m s$^{-1}$ for the strong wind class. The predominately positive wind speed increments throughout the UTLS represent a wind increase





which is largest in the 500 m layer below LRT$_{yo}$ ranging between 0.1 m s$^{-1}$ for the weak and 0.45 m s$^{-1}$ for the strongest wind
class. The positive residuals, show that a slow wind speed remains in the analysis, however the weaker residuals than the innovations observed for each class point to an improvement of wind. For the strongest winds the innovations are reduced by up to 50 %.

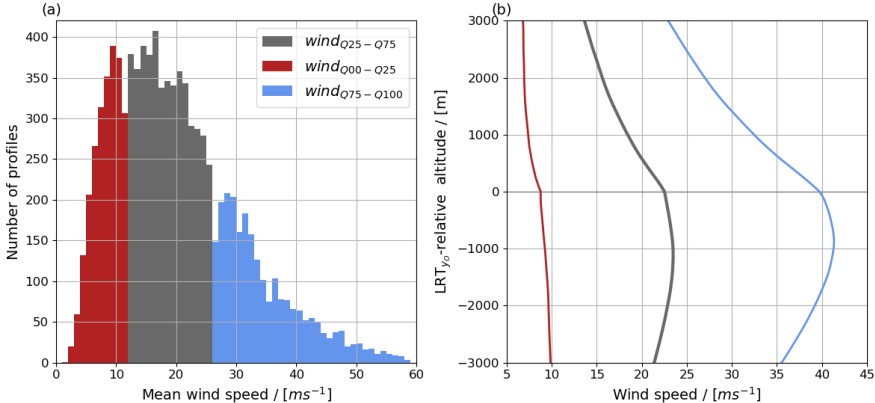

**Figure 9**: (a) Distribution of mean observed wind speed in the ±3 km above and below LRT$_{yO}$ with 1 m s$^{-1}$ bin size. Quartiles of the
distribution in (a) are used to distinguish wind classes: weakest wind (red; Q00–Q25), intermediate wind (grey; Q25–Q75) and strongest wind (blue; Q75–Q100). (b) The corresponding LRT$_{yo}$–relative mean profiles of observed wind speed per class.

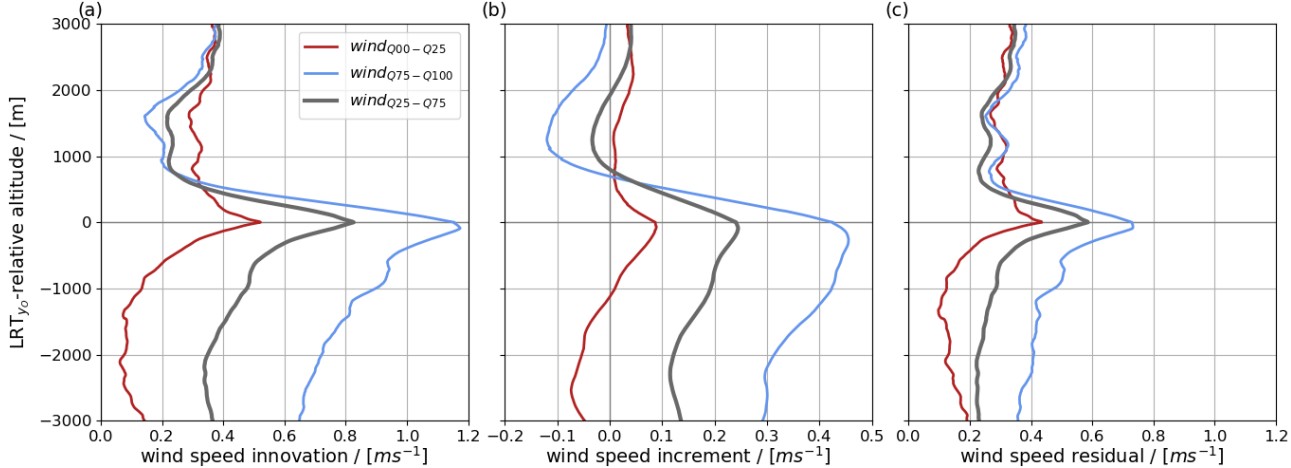

**Figure 10:** LRTyO–relative mean profiles of wind speed (a) innovations, (b) increments and (c) residuals per classes defined in Fig. 9.

### 3.3 Influence on the tropopause altitude

In the following, we investigate how the modification of the temperature profile (Fig. 5e) affects the tropopause altitude. In the subsequent, LRT altitude differences between the observations and the background (LRT$_{yO}$–LRT$_{yFG}$) are referred to as "LRT innovations" according to Eq. (1). LRT altitude differences between the observations and the analyses are referred to as "LRT residuals" (LRT$_{yO}$–LRT$_{yAN}$; Eq. 2), respectively. An overview of LRT innovations for the entire data set is given in Fig.



11a providing a symmetric normal distribution centered near zero (–26 m). For 43 % of the profiles, the LRT innovations are
in range of ±100 m. For about 10 % of the profiles, LRT differences are larger than 1 km. In order to prevent the impact of
LRT misdetections (see discussion in Sect. 3.2.2), which are most likely for unusually large LRT differences, the following
evaluation is restricted to 8778 profiles (~90 %) which provide LRT altitude innovations within ±1 km.

First, we compare the LRT innovations (Fig. 11b) and LRT residuals (Fig. 11c): In Fig. 11b (subset of Fig. 11a) different
intervals of LRT innovations are colour-coded. The grey interval reflects LRT innovations within ±100 m, while bluish colours
represent profiles where the observed LRT is higher than the background and reddish colours vice versa. The colouring of Fig.
11b is preserved in Fig. 11c to identify whether profiles have changed the interval. Only a small fraction of profiles exhibits
an $LRT_{yAN}$ vertically closer or further away from the $LRT_{yo}$ with respect to the classes of LRT innovations. The distribution
of LRT innovations shows a clear maximum near zero with a frequency corresponding to about 3000 profiles per bin, and the
frequency decreases towards the edges of the distribution to ~10 profiles per bin. The distribution of LRT residuals (Fig. 11c)
shows a slightly increased number (+15 %) of profiles within ±100 m indicating an improved tropopause altitude in the
analysis. This is also confirmed by a slightly narrower shape of the gaussian fit of the LRT residuals compared to the LRT
innovations.

For the different intervals of LRT innovations in Fig. 11, Table 1 provides number of profiles, the mean $LRT_{yO}$ altitude,
innovation, residual and improvements. Except for the interval with the smallest innovation (±100 m; grey interval), the
average innovation is larger than the residual which implies a vertical shift of $LRT_{yFG}$ towards $LRT_{yO}$. The generally positive
influence is supported by the depicted improvements, defined as the absolute difference of innovation and residual.
Interestingly, both, the LRT altitude shift and also the improvement grow with increasing distance between $LRT_{yO}$ and $LRT_{yFG}$.
For the interval of smallest innovations, Table 1 shows a slightly higher average LRT residual compared to the innovation,
which is a result of a small number of profiles with deteriorated tropopause altitude.

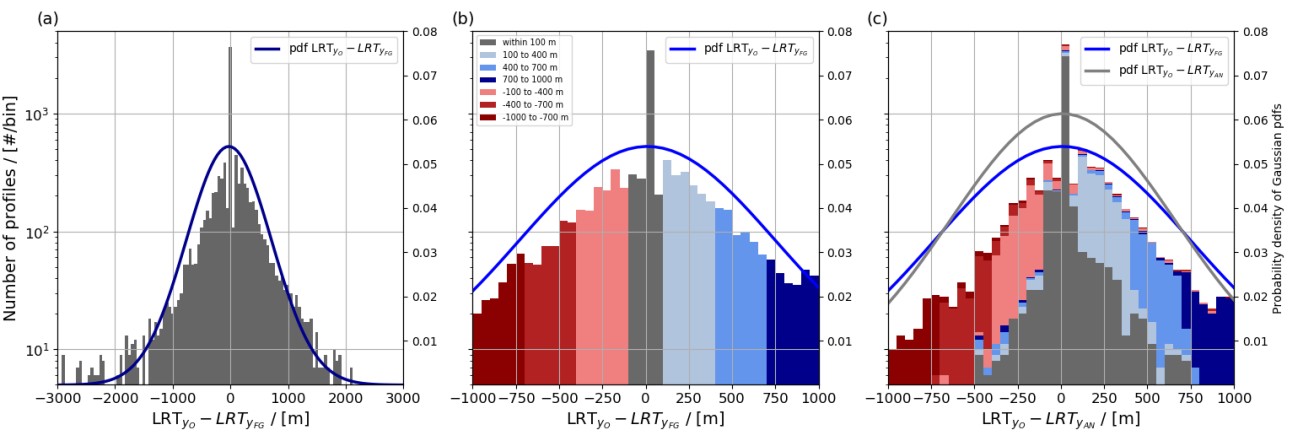

**Figure 11:** Distribution of (a) LRT innovations in the full data set, (b) as in (a) but for the range ±1000 m and (c) LRT residuals. The colour
coding reflects intervals of LRT innovations shown in (b) and is reused in (c) to visualize the LRT altitude change in $LRT_{yAN}$ (for details
see text). Gaussian probability density function (pdf) are given in lines of dark blue, blue and grey, respectively, for 50 m bins. Note the log-
scale of the y-axis.



| LRT innovation intervals [m] | Number [#] | LRT$_{yo}$ [m] | Innovation [m] | Residual [m] | Improvement [m] |
|---|---|---|---|---|---|
| All profiles | 9729 | 12226 | –26 | 12 | 79 |
| –1000 to –700 | 209 | 12806 | –812 | –420 | 371 |
| – 700 to –400 | 473 | 12425 | –525 | –277 | 227 |
| – 400 to –100 | 1309 | 11963 | –237 | –121 | 76 |
| – 100 to 100 | 4196 | 12064 | 2 | 18 | –33 |
| + 100 to 400 | 1637 | 11706 | 222 | 175 | 36 |
| + 400 to 700 | 627 | 12456 | 517 | 349 | 153 |
| + 700 to 1000 | 227 | 13607 | 830 | 555 | 271 |

**Table 1:** Number of profiles, averaged observed LRT altitude, innovation, residual and improvement for different intervals of LRT innovation (see Fig.11 b). The improvement is defined as the averaged $|LRT_{yO}-LRT_{yFG}|-|LRT_{yO}-LRT_{yAN}|$, so positive values reflects an improved LRT altitude in the analysis.

In order to understand how the LRT altitude changes are related to the changes of the background temperature profile, the average temperature increments for the individual intervals of LRT innovations are presented with respect to $LRT_{yFG}$-relative

altitude (Fig. 12). For small LRT innovations (within ±100 m, grey line in Fig. 12 a,b), the temperature increments are negative at the LRT (–0.15 K) and positive in the LS (0.2 K) (analogous to the sharpening influence discussed in Sect. 3.1 and Sect. 3.2) which does not lead to major changes in the LRT altitude in this interval (see Tab. 1). With increasing LRT innovations, the altitudes of the peaks in the $LRT_{yFG}$–relative increments are vertically shifted (compare coloured profiles in Fig. 12). In case of the negative LRT innovations (Fig. 12a), which means that the observed $LRT_{yO}$ is located lower than the background

$LRT_{yFG}$, we observe positive increments (warming, 0.3–0.6 K) at and above $LRT_{yFG}$ and negative increments (cooling, 0.2– 0.4 K) below $LRT_{yFG}$. As the strongest negative increments are located near the observed tropopause (dotted lines in Fig. 12) peaks in the increments are shifted downwards and show slightly higher maxima for more negative LRT innovations (red profiles in Fig. 12a). In contrast, positive LRT innovations (i.e., $LRT_{yO}$ located above $LRT_{yFG}$; Fig. 12b) exhibit negative increments (–0.3 to –0.4 K) above $LRT_{yFG}$, and positive increments below the $LRT_{yFG}$. Here, the increment peaks are shifted

upwards for more positive LRT innovations (blue profiles in Fig. 12b). The peaks of the negative increments coincide with the altitude of the observed tropopause. This confirms the indication of Fig. 4 that the negative (positive) increments are mostly aligned to (located above) the observed tropopause altitude. In summary, we found a positive influence of DA on the tropopause altitude, which is expressed by the smaller LRT residuals than LRT innovations. The potential relationship between this LRT altitude improvement and the vertically shifted temperature increments is discussed in more detail in Sect. 4.



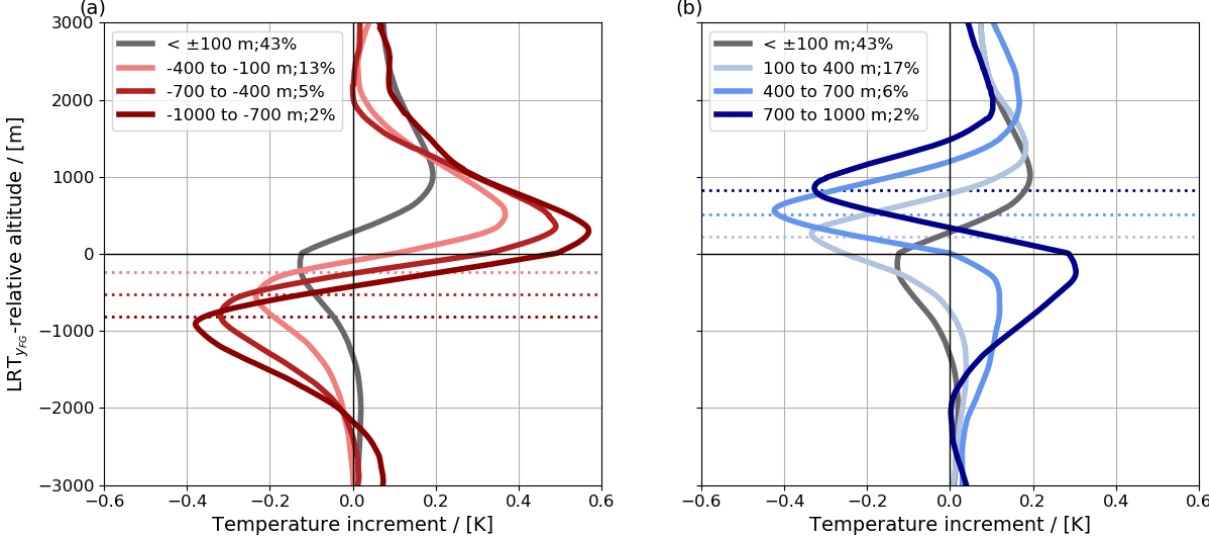

**Figure 12:** Mean temperature increments with respect to $LRT_{yFG}$–relative altitude for intervals of LRT innovations (colour coding, following Fig. 11). (a) Negative LRT innovations ($LRT_{yo} < LRT_{yFG}$, in red) and (b) positive LRT innovations ($LRT_{yo} > LRT_{yFG}$, in blue). The grey lines show the increment for LRT innovations within ±100 m. In (a) and (b) the averaged $LRT_{yFG}$–relative altitude of $LRT_{yo}$ for each interval is depicted by the dotted lines.

### 3.4 Attributing the influence to the radiosondes

The presented results revealed that DA sharpen the tropopause at the location of the radiosondes, which provides a strong indication that this influence is related to the information contained in the radiosondes. However, a potential contribution of other observations cannot be excluded. For this reason, we compare the profiles and increments of the 497 NAWDEX radiosondes in the CTR run with the DEN run, in which they are denied and only passively monitored. The average profiles of observed temperature and N² of the 497 NAWDEX profiles (Fig. 13) are comparable to the average profiles of the 9729 radiosondes (Fig. 5) with a similar magnitude of the N² jump of N² and an alike decrease of wind shear across the tropopause. The average minimum temperature at the tropopause (Fig. 13a) is slightly higher (by ~2 K) which is related the observation locations of the NAWDEX radiosondes at higher latitudes where the tropopause is typically lower and warmer. The additional radiosondes show a pronounced wind maximum below $LRT_{yo}$ of about 29 m s$^{-1}$ (Fig. 13c), which compared to the lower wind speeds in the complete data set (Fig. 5c), indicates the occasionally strong jet streams in the focus of the NAWDEX campaign (Schäfler et al., 2018).

The increments for temperature, N², wind speed and wind shear for the subset of 497 additional NAWDEX radiosondes are presented in Figures 13e-f. The CTR run exhibits a vertical structure that is comparable to the complete data set as discussed in Sect. 3.2: Temperature increments are negative (–0.25 K) around the observed tropopause, and positive in the LS (1–2 km above $LRT_{yO}$). Accordingly, the CTR N² increments possess a similar distribution with a 1 km layer of positive increments just above $LRT_{yO}$ with a maximum of 0.3x10$^{-4}$ s$^{-2}$, weak negative increments (-0.2x10$^{-4}$ s$^{-2}$) in a 1 km layer beneath $LRT_{yO}$



and increments around $\pm0.1\times10^{-4}$ s$^{-2}$ beyond the 1 km layers. The CTR vertical structure of wind speed increments for the NAWDEX radiosondes also agree with the complete data set (c.f. increments for the strongest wind class, Fig. 10b), with a positive increment (~0.2 m s$^{-1}$) in the UT and a negative increment in the LS. Wind shear increments in the CTR run are also

similar, but the minimum wind shear just above the LRT$_{yO}$ shows slightly lower values ($0.5\times10^{-3}$ s$^{-1}$) compared to Fig. 5h. The increments of temperature, N², wind speed and shear in the DEN run (Fig. 13) are weaker at each altitude and tend to pick up a similar vertical distribution. This implies that the main contribution of the tropopause sharpening and influence on wind comes from the assimilated radiosondes, but the non–zero DEN increments indicate other observations to influence the tropopause structure in the same direction. This may be due to either the remote impact of operational radiosondes or other

vertically resolved observations as e.g., GPS radio occultation or dropsonde observations.

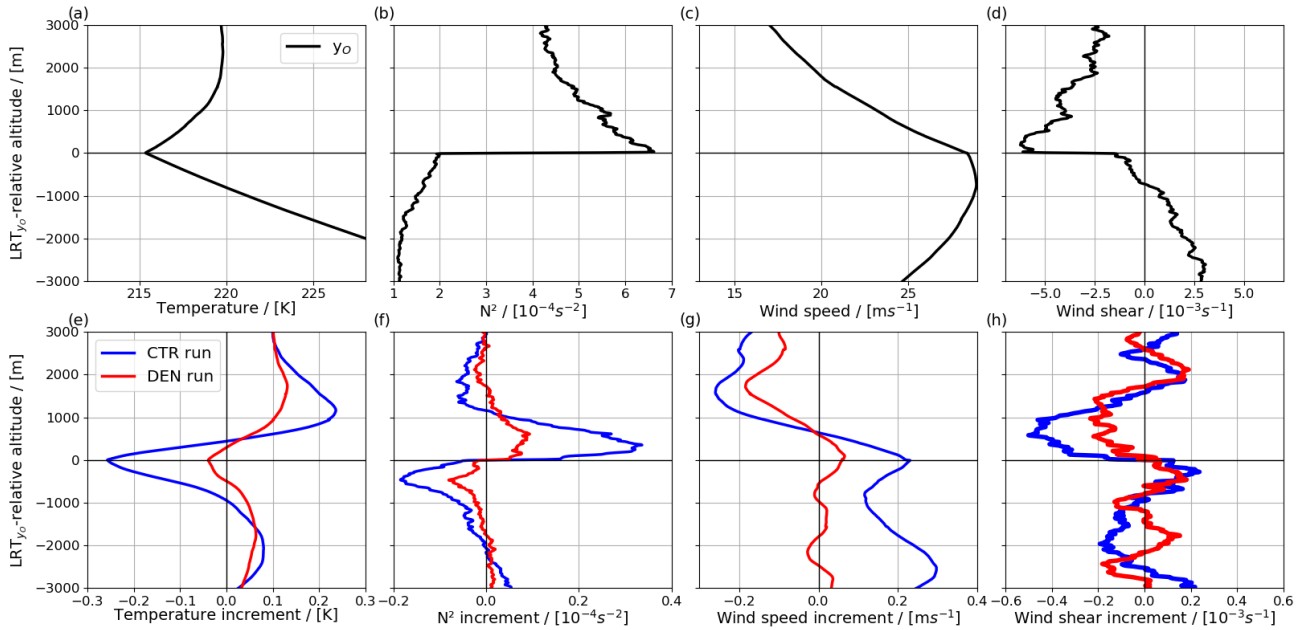

**Figure 13:** LRT$_{yO}$–relative mean profiles of the (a) temperature, (b) N², (c) wind speed and (d) wind shear as observed by the 497 NAWDEX radiosondes and the respective increments of (e) temperature, (f) N² and (g) wind speed and (h) wind shear for the CTR (blue) and DEN (red) experiment.

**4 Discussion**

In this study we evaluate the influence of DA on the structure of the tropopause in the ECMWF IFS based on 9729 midlatitude radiosonde profiles. The statistical evaluation of observed temperature and wind as well as derived N² and wind shear in (thermal) tropopause–relative coordinates reproduces the typical sharp vertical gradients at the midlatitude tropopause (Birner et al., 2002). The LRT altitudes between 6–18 km during fall are considered to be representative for this area and season (e.g.,

Krüger et al., 2022).





To address the influence of DA on tropopause sharpness and altitude, the radiosonde and model states are transferred to tropopause–relative coordinates. The selection of a suitable reference is challenging because the tropopause–relative distributions in observations, background and analysis vary in the different LRT–relative coordinates due to slightly varying individual tropopause altitudes. Such LRT altitude differences may result from either mis–detections caused by slight

temperature fluctuations in the upper troposphere or from differences in the 3D temperature distribution, e.g., in the vicinity of the jet streams. Since the origin of these LRT altitude differences cannot clearly be identified and they affect the evaluation of the tropopause altitude influence, we only consider LRT differences in the range of ±1 km. The sharpest tropopause in the observations, background and the analysis occur when viewed with respect to the "own" LRT. However, it turned out that the impact on sharpness cannot be assessed in own LRT coordinates as LRT altitude deviations in the background and analysis

profiles cause a spurious tropospheric temperature bias. The observed tropopause as a reference provides the most accurate representation of the tropopause to evaluate the influence on tropopause sharpness. However, temperature increments with respect to the background tropopause are useful to better understand changes to the background profile. This shows that the tropopause reference system needs to be purposefully and carefully selected and uncertainties related to the tropopause detection need to be considered.

In this study, we highlight that DA improves the underestimated sharpness of the tropopause by introducing systematic changes to the background temperature and wind profile. The temperature innovations indicate a warm bias at the tropopause and a cold bias in the LS. In this layer of the sharp reversal in thermal stratification, $N^2$ is overestimated (underestimated) by the background below (above) the tropopause confirming findings by Birner et al. (2002). The temperature increments tend to move the background temperature towards the observations by decreasing the tropopause temperature by ~0.25 K and

increasing it with a similar magnitude in the LS above, which was already indicated by Radnoti et al. (2010) for the UTLS. The accompanied increase of $N^2$ ($0.3 \times 10^{-4}$ s$^{-2}$) in a 1 km layer above the tropopause and a decrease of $N^2$ (~$0.2 \times 10^{-4}$ s$^{-2}$) in the uppermost troposphere is equivalent to a tropopause sharpening, which is consistent in shape and magnitude with Pilch Kedziersky et al. (2016). Although the increments clearly reduce the background biases, the influence is rather small (~10 %) compared to the innovations. The remaining LS cold bias in the analysis (0.2 K) corresponds to previous assessments (Radnoti

et al., 2010; Shepherd et al., 2018; Bland et al., 2021). The warm bias at the tropopause (1.2 K) is in line with Ingleby et al. (2016). However, the magnitude of the warm bias is roughly 2–3 times smaller compared to Bland et al. (2021). This difference may be related to vertical smoothing of the radiosonde profiles in Bland et al. (2021), which could lead to a higher tropopause temperature (König et al., 2019). Large $N^2$ biases ($-2$ to $3 \times 10^{-4}$ s$^{-2}$) in the analysis are found in the ±0.5 km layer around the tropopause. In addition, we show that the magnitude of the tropopause sharpening depends on the dynamic situation. For

sharper tropopauses, which are typically related to higher and thus colder tropopauses occurring in situation of upper–level ridges (Pilch Kedziersky et al., 2015), temperature (and thus $N^2$) increments, innovations and residuals are larger.

Positive wind innovations (~about 1 m s$^{-1}$ near the tropopause) reveal the existence of a slow wind bias in the background, particularly for the wind maximum, which confirms findings by Schäfler et al. (2020) and Lavers et al. (2023). In the layer of ± 1 km around the tropopause the observations show a sharp contrast of vertical wind shear from positive shear in the up to



the wind maximum in the UT to negative shear in the LS, with the strongest gradient of shear located at the tropopause. We find positive wind increments in the UT with a peak at the tropopause (0.2 m s$^{-1}$) leading to a corresponding acceleration of wind speed and nearly unchanged wind in the LS. Wind shear is underestimated by the background in the uppermost troposphere and overestimated in the lowermost stratosphere. The wind increments are associated by positive shear increments just below the tropopause and negative shear increments in a 1 km layer above the tropopause. The generally positive influence

of DA at all altitudes on the wind is depicted by the smaller analysis than background bias for different wind speeds. However, we find that high wind speed situations are characterized by increased bias in the background around the tropopause (underestimation of 1.2 m s$^{-1}$) which is reduced by almost 50 % in the analysis. This confirms Schäfler et al. (2020) who speculated that large wind errors near the jet stream in IFS short–range forecasts are reduced in the analysis. The stronger positive impact on wind for high wind speed situations was recently demonstrated by Lavers et al. (2023).

In a further investigation, we found that the influence on the temperature profiles also affects the vertical position of the tropopause altitude in the analysis. While for individual profiles the LRT altitude difference of observations, background and analysis can exceed 1 km, the average differences are small (< 50 m) compared to the vertical resolution of the model of about 300 m at the tropopause. Bland et al. (2021) showed a higher tropopause altitude of about 200 m in IFS analyses using a previous model cycle (Cy41r2) and 3204 radiosondes that are a subset of the data set analysed in this study. Again, the differing

results may be related to the vertical smoothing of the radiosonde profiles (König et al., 2019).

We reveal a positive influence of DA on the representation of the LRT altitude in the analysis that is closer to the observations than the background (or first–guess). In case the LRT altitude of background and observation is comparable (in the range of ±100 m) the analysis tropopause altitude shows a slightly higher distance to the observed tropopause (16 m), which is small compared to the vertical resolution of the IFS. In case of increased tropopause differences (LRT innovations >100 m), the

analysis shows a systematically improved LRT altitude whereas the improvement grows with increasing LRT innovation. The vertical shift of the temperature increments with respect to the background tropopause agrees with the resulting LRT altitude changes in the analysis: If the observed tropopause lies below the background tropopause, the region below is cooled, which leads to a lower LRT in the analysis. In contrast, if the observed LRT is located above the background, the region above is cooled, which, on average, shifts the analysis LRT upwards. Bland et al. (2021) and Schmidt et al. (2010) show that local

temperature changes in the UTLS affect the tropopause altitude. For instance, a cooling of the LS and a warming of the UT leads to a higher tropopause altitude in models. The opposite effect, i.e. a lower located tropopause, is true in case of a cooling of the UT and a warming of the LS. The changes of the temperature observed in this study that are induced by the DA thus provide a reasonable explanation for the changed representation of the tropopause altitude.

The analysis of a subset of 497 NAWDEX profiles considered in a data denial OSE allowed the sharpening to be attributed

directly to the assimilation of the radiosondes. The control run increments which assimilated NAWDEX radiosondes, showed a similar shape and magnitude as the full data set. The increments in the denial run, where the non–operational radiosondes were only passively monitored, are much weaker, but the positive and negative increments pointing in the same direction as the control run. Hence, the radiosonde assimilation provides the major contribution to the increments (and thus the sharpening



influence), which likely holds for the entire data set of the presented results. The non–zero increments in the denial run might
be related to the assimilation of other observations, for instance GPS–RO data (Pilch Kedziersky et al., 2016), or to the
contribution of the routinely radiosondes that are assimilated in the same assimilation time–window at a close–by location. A
more sophisticated OSE with more observations and different observation types to be denied would be required for a deeper
investigation of this effect. The approach of assessing an OSE in observation space allows to evaluate the influence of the
observations on temperature and wind distributions on a local scale. However, the $\boldsymbol{B}$–matrix in hybrid 4DVAR schemes spreads
information of assimilated observations horizontally and vertically in space and time. This poses the question to which extent
the sharpening influence on the temperature and wind gradients in the UTLS, but also on the tropopause, affects not only
locally but also the surrounding region in the model. To answer this question, the authors work on an evaluation in model
space in a subsequent study.

**5 Conclusion**

Weather and climate predictions strongly rely on an accurate representation of the sharp cross-tropopause gradients of
temperature and wind. However, the initial conditions of current NWP models substantially underestimate these gradients, i.e.,
the sharpness of the tropopause. DA is known to correct for erroneous vertical distributions of temperature and wind in the
model background forecast. In this study, we address the question whether DA (positively) influences the sharpness and
altitude of the midlatitude tropopause. For this purpose, a large data set of radiosonde observations observed during a one–
month period in fall 2016 is compared with ECMWF IFS background and analysis profiles. The main conclusions of this study
following the research questions raised in the Introduction are summarized below:

*1. How is tropopause sharpness represented in background forecasts and what is the influence of DA on the analysis? Does
the diagnosed temperature and wind influence depend on the tropopause structure and vary in different dynamic situations?*
The tropopause–relative analysis of the DA influence on temperature, $N^2$, wind speed and shear using the 9729 radiosondes
shows that the tropopause is sharpened. This sharpening is described by an average cooling at the tropopause (0.25 K) and a
heating (0.25 K) of the LS (0.5 to 1.5 km above the observed tropopause). These increments are corresponding to an increase
of $N^2$ ($0.3 \times 10^{-4}$ s$^{-2}$) in a 1 km layer just above the tropopause. We furthermore find an acceleration of wind speed (~0.2 m s$^{-1}$)
which is most pronounced at the altitude of the highest observed wind speeds. The sharp contrast of wind shear from positive
values in the UT to negative values at the LRT and in the lowermost LS is increased. For each parameter, the increments
sharpen the tropopause, however, the influence is found to be small compared to the magnitude of the model background
biases. We further uncover a sensitivity of the influence to different dynamic situations. Larger increments, but also larger
innovations/residuals, are connected to sharper ($N^2_{max}$ used as indicator) tropopauses, that are typically associated with ridge
situations (high tropopause), while a weaker influence is observed for smoother classified tropopauses, which are related to
troughs. In addition, we investigated the influence on the cross–tropopause wind distribution, and detect reduction of the slow



wind bias across the tropopause. For strong jet stream wind situations, we find the largest influence corresponding to a reduction of the bias by ~50 % in the UT.

*2. Does the influence on the temperature profile affect the tropopause altitude?*

A unique aspect of this study is that the DA influence on the tropopause altitude is systematically addressed. On average, tropopause altitudes differences of observations to the background and the analysis are within 50 m and for about 90 % of the profiles the tropopause altitudes agree within ±1 km. We find a positive influence of DA on the tropopause altitude in the analysis, which is expressed by a narrower distribution of the LRT residuals compared to the LRT innovations. With increasing difference between observed and background tropopause, we detect a stronger positive influence on the tropopause altitude in

the analysis. The altitude improvement can be attributed to systematic temperature increments relative to the background tropopause which cause a distinct vertical shift depending on the position of the observed tropopause. If the background tropopause is located either higher or lower than the observed, the temperature increments pull the background towards the observed tropopause.

*3. Can the diagnosed influence be attributed to the assimilated radiosondes or do other observations also affect the tropopause structure?*

The comparison of increments for 497 non-operationally launched radiosondes within an OSE confirms that the diagnosed influence (sharpness and altitude of the tropopause, wind acceleration in UT) can be mainly attributed to the assimilated radiosondes. However, the non–zero increments in the run without the NAWDEX radiosondes reveal that other observations

also contributed to the sharpening and to the increase of wind at the radiosonde locations. The novel approach of a tropopause–relative assessment in observations space combined with an OSE complements previous studies by providing a novel perspective on the local influence of DA on the tropopause that allows a positive influence to be assigned to the assimilation of radiosonde observations. Although the influence on the temperature and wind profiles is found to be small compared to the background and analysis errors, DA is able to improve the sharp gradients of temperature and wind at the tropopause. The

increased vertical gradients of temperature and wind are expected to improve the tropopause PV distribution (as indicated in Lavers et al., 2023). The sharpening process likely counteracts the decreasing forecast PV gradients. Future increases of horizontal and vertical model resolution in NWP and improved parameterizations of processes that modify the tropopause sharpness may positively impact the representation of the tropopause structure and thus the quality of NWP.

**Data availability**

The feedback files analysed in this study are requested from via the Meteorological Archival and Retrieval System (MARS) documented at https://confluence.ecmwf.int/display/UDOC/MARS+user+documentation, last access: 1st June 2023. We are grateful to the ECMWF for granting the data access.



**Author contributions**

KK performed the data analysis, produced the figures and wrote the manuscript. AS retrieved the observation feedback files.
KK, AS, MW and GC conceptualized the study. MW and GC gave important guidance for the study, helped with the interpretation of the results and commented on the paper.

**Competing interests**

The authors declare that they have no conflicts of interest.

**Acknowledgements**

We thank the European Meteorological Service Network (EUMETNET), Environment and Climate Change Canada (ECCC) the Icelandic Meteorological Office (IMO) and DLR for supporting NAWDEX with additional radiosonde launches. The authors thank Gabor Radnóti (ECMWF) for his technical assistance and set–up of the OSE. We are particularly grateful to the ECMWF for data access and allocation of computational resources. The research leading to these results has been done within the subproject "A3" of the Transregional Collaborative Research Center SFB / TRR 165 "Waves to Weather"
(www.wavestoweather.de) funded by the German Research Foundation (DFG)." KK thanks Waves to Weather for the financial support of a 11 week stay in the Numerical Weather Prediction and Data Assimilation group of MW at the Department of Meteorology and Geophysics of the University of Vienna. Representative for the whole department, we particularly thank Tobias Necker, Philipp Griewank, Andreas Stohl, Leopold Haimberger, Stefano Serafin and Lukas Kugler for their valuable comments on the results of this study. We thank Sonja Giesinger (DLR) for the careful review of the manuscript.

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
