# Peer review of "Influence of radiosonde observations on the sharpness and altitude of the midlatitude tropopause in the ECMWF IFS"

_EGUsphere, 2023_

## Author Comment (AC1)

Dear reviewer,

we are glad about your positive impression of our manuscript and that you consider it interesting and worth to be published. Your valuable comments and specific suggestions helped us to improve the manuscript.

Below, we answer each of your comments using a blue font. We also uploaded a revised version (blue and italic) that includes a few additional technical corrections using track changes.

**General comments**

I found the manuscript to be useful and interesting. It should be published after revision. It is generally well written, but a little long in places. The description of the biases round line 475 needs improvement, see detailed comments.

Boer, G. J. 1983 Homogeneous and isotropic turbulence on the sphere. J. Atmos. Sci., 40, 154-163

Daniel Hodyss, Nancy Nichols, The error of representation: basic understanding, Tellus A, 2015, 67, 0

Janjic, T., N. Bohrmann, M. Bocquet, J.A. Carton, S.E. Cohn, S.L. Dance, S.N. Losa, N.K. Nichols, R. Potthast, J.A. Waller, and P. Weston, 2018: On the representation error in data assimilation. Q.J.R. Meteorol. Soc. 144, 1257-1278, https://doi.org/10.1002/qj.3130.

General comment I)

A possible (hopefully minor) addition: is there a relation between the sharpness of the temperature inversion and the maximum wind speed? (Perhaps one should look at this separately for mid-latitudes and sub-tropics?)

This is an interesting question. We first address the separated consideration of mid-latitudes and sub-tropics, which is in line with a suggestion by the second reviewer. Based on the observed tropopause altitudes in Fig. 2 we define profiles with an LRT >14 km as sub-tropical (25 % of the profiles) and LRT <14 km as mid-latitude profiles (75 %). The tropopause-relative profiles of observed temperature, $N^2$, wind and wind shear as well as the increments are shown for both classes in Figure S2. Compared to the mean midlatitude profiles which shows similar distributions as for the overall data set (compare Fig. 5). The sub-tropical profiles exhibit a considerably lower temperature in the entire UTLS, a weaker LS temperature inversion, a cooler tropopause and furthermore show continuously decreasing wind speed with altitude and no wind maximum being located near the tropopause. This represents a typical temperature and wind distributions one might expect poleward of the sub-tropical jet. We consider this an interesting finding relevant for the reader and decided to

add Fig. S2 to the Supplement. In addition, we added the following description in Sect. 2.2 (p.9, ll.226-228 in the revised version):

*"A separate consideration of extratropical (LRT < 14 km) and sub-tropical (LRT > 14 km) observations reveals similar shapes for the extratropical and the overall data (see Fig. S2a-d). The sub-tropical mean profiles exhibit lower temperatures in the entire UTLS, a weaker temperature inversion in the LS and no wind maximum being located near the tropopause."*

[Figure]

Figure S2: $LRT_{yO}$–relative mean profiles of (a) temperature, (b) $N^2$, (c) wind speed, (d) wind shear for profiles associated with the mid-latitudes ($LRT_{yO}$ < 14 km; solid) sub-tropics ($LRT_{yO}$ > 14 km; dashed).

In addition, we added a plot for the average increments (Fig. S3). Increments of sub-tropical profiles are weaker, but still point in the same direction as in the midlatitudes. The wind speed increments are smaller in the upper troposphere at lower wind speeds. We added Fig. S3 to the Supplement and a description to Sect 3.2 (p.11, ll. 270-272 in the revised version).

*"A separate analysis of mid-latitude and sub-tropical increments (Fig. S3) shows that the latter are weaker. However, as the increments in both regions point in the same direction, the complete data is considered for the statistical analysis in the remainder of this article."*

[Figure]

Figure S3: as in Fig. S2 but for increments.

Next, we evaluate the relation between the tropopause sharpness and maximum wind speed. We show the mid-latitude wind profiles for different classes of tropopause sharpness (following the analysis in Sec 3.2.3 Fig. 7). The profiles with the sharpest tropopause (blue) have slightly stronger average winds in the UT and weaker winds in the LS.

[Figure]

Figure: LRT$_{yO}$–relative mean profiles of (a) wind speed and (b) per classes of tropopause sharpness for mid-latitude profiles with LRT < 14 km.

A few remarks on the distribution of wind speed relative to the LRT and the dependence of maximum wind and tropopause sharpness: The maximum wind speed 1-2 km below the tropopause is related to the typical distribution of winds and LRT in the midlatitudes (see Birner et al., 2002), which we try to illustrate using a figure (see below) taken from Krüger et al. (2022, ACP). The figure shows a meridional transect through the polar jet stream (magenta contours) with the typical LRT (thick dotted black line) discontinuity at the jet stream (e.g., Pan et al., 2004). The maximum wind speeds occur below the elevated tropopause, which agrees with the diagnosed maximum winds. We do not expect a strong relationship between tropopause sharpness (defined by N²$_{max}$) and the jet stream wind speeds in general. Typically, sharpest tropopauses occur away from the jet stream (Gettelman et al., 2011). Accordingly, we think that the higher winds for sharper tropopauses are related to the sharper and higher tropopause altitudes on the southern side of the jet stream. Wind speeds in the jet are rather related to isentropic gradients of potential vorticity (e.g. Bukenberger et al., 2023, see also the dynamical tropopause (2 PVU, thin black line) that ascends almost vertically through the jet stream).

[Figure]

Vertical cross sections potential temperature (grey contours), the isopleths of the wind speed (magenta contours), and the thermal (thick black dots) and the dynamical tropopause (2 PVU, black isoline). Colours show ERA5 humidity bias (colour shading). Figure taken from Krüger et al. (2022, ACP).

References:

Bukenberger, M., Rüdisühli, S., and Schemm, S.: Jet stream dynamics from a potential vorticity gradient perspective: The method and its application to a kilometre-scale simulation. Q. J. Roy. Met. Soc., 149, 2409–2432, https://doi.org/10.1002/qj.4513.

Gettelman, A., Hoor, P., Pan, L. L., Randel, W. J., Hegglin, M. I., and Birner, T.: The extratropical upper troposphere and lower stratosphere, Rev. Geophys., 49, RG3003, https://doi.org/10.1029/2011RG000355, 2011.

Krüger, K., Schäfler, A., Wirth, M., Weissmann, M., and Craig, G. C.: Vertical structure of the lower-stratospheric moist bias in the ERA5 reanalysis and its connection to mixing processes, Atmos. Chem. Phys., 22, 15559–15577, https://doi.org/10.5194/acp-22-15559-2022, 2022.

Pan, L. L., Randel, W. J., Gary, B. L., Mahoney, M. J., and Hintsa, E. J.: Definitions and sharpness of the extratropical tropopause: A trace gas perspective, J. Geophys. Res.-Atmos., 109, D23103, https://doi.org/10.1029/2004JD004982, 2004.

General comment II)

On a related note it would be interesting to define a set of 300 m vertical layers (similar resolution to the IFS near the tropopause) average high resolution observed profiles over these layers and see how the averaged profiles compare with the original ones.

Some properties such as 'Profiles with sharper tropopause exhibit stronger background biases' are more-or-less inevitable when averaging onto a coarser grid.

First of all, we want to clarify that our analysis did not involve any radiosonde raw data. Such data, as also referred to in a later comment, would have a vertical resolution of ~5 m. In our study we access the radiosonde data through the data assimilation output which is provided in the monitoring feedback files. These files contain the quality-controlled and thinned observation profiles (and their model equivalents) as assimilated by the ECMWF. The number of available levels per profile (see Fig. S1a) exhibits a bi-modal distribution, which is related to a mix of different assimilated radiosonde report types (low resolution alphanumeric TEMP and high resolution BUFR) available for assimilation (see also Ingleby et al., 2016). The vertical resolution (distance between two neighboring data points) in the UTLS (Fig. S1b) also shows a bi-modal distribution, with the modes corresponding to the different report types. As we think this information helps to better understand the results we added Fig. S1 to the Supplement and revised the description in Sec.2.1. Please note that we also added information about how many profiles are low- and high resolution (see your comment on ll110-113 below):

[Figure]

Figure S1: Histogram of (a) the number of vertical levels (b) the derived average vertical distance between two neighboured levels of the radiosonde observations as provided by the requested feedback files.

In addition, we tested the influence of a reduced vertical resolution (see figure below). As expected, the vertically smoothed profile (300 m red dotted line) shows a warmer tropopause (~0.2 K) compared to the profiles interpolated at the 10 m resolution (black solid). This effect of smoothing on the temperature profile is small and likely a consequence of the vertical resolution of the assimilated data (100-400 m) in the UTLS (+- 3 km around the LRT) being close to the vertical resolution of the model (300 m). Although such smoothing -if applied to radiosonde raw dataweakens tropopause sharpness (as for instance demonstrated in Koenig et al., 2019) it would not significantly change the results of this paper. We used the data at highest resolution in order to guarantee a reliable tropopause altitude detection and also as we are interested how far the model is from the provided "nature" within the observations.

König et al., 2019.: Tropopause altitude determination from temperature profile measurements of reduced vertical resolution, Atmos. Meas. Tech., 12, 4113–4129, https://doi.org/10.5194/amt–12–4113–2019, 2019.

[Figure]

Figure: $LRT_{yO}$–relative mean temperature for the profiles as given in the preprint (vertically interpolated to a 10 m grid; black solid line) and as averaged over 300 m vertical bins (red dashed line).

General comment III)

In the data assimilation literature 'representation error' (or representativeness errors) is a useful concept relevant to this manuscript (see e.g. Hodyss and Nichols, 2015; Janjic et al, 2018). For wind I find it useful to think of the spectra of rotational and divergent wind (eg Boer, 1983).

Any finite numerical model truncates the spectra and hence reduces the wind speed (only slightly in general but more where there is a lot of variability on small scales, e.g. near a jet stream).

We agree with the reviewer that this comparison is certainly affected by representativeness errors when comparing point measurements to grid-average NWP values as described in e.g., Janjic et al., 2017. This effect can be addressed partly through averaging of the profiles to a comparable vertical resolution, which we discussed in the previous comment. However, the horizontal resolution and the role of horizontal gradients cannot be investigated (which would be also important to address the representativeness error). Validation studies of jet stream winds (Schäfler et al. 2020) and UTLS humidity (Krüger et al. 2022) based on two-dimensional lidar cross-sections (see figure above) showed large horizontal and vertical scales with coherent error structures (often several hundred kilometres horizontally and 1–2 km vertically) which should be represented on the grids used by the NWP models. We added three sentences to the discussion (see revised version p.22, ll.518-522)

*"In this study, profiles were interpolated to a 10 m vertical grid to guarantee an accurate detection of the LRT. Certainly, the comparison is affected by representativeness errors when comparing point measurements to grid-average NWP values as discussed (Weissmann et al., 2005; Hodyss and Nichols, 2015; Janjic et al, 2018). Such an effect could be partially addressed through vertical averaging of the profiles, however, the vertical resolution of the assimilated data (100-400m, see Fig. S1) is already close to the model grid spacing in the UTLS (~300 m)."*

**Specific/technical comments**

===========================

Abstract - a bit long

*As we could not find an official length limitation and want to provide a complete summary of our findings. We nevertheless made a few corrections in the abstract to make it shorter and clearer.*

lines 35-36 'Above the tropopause ...' make it a bit clearer that this is a description of average or typical conditions.

*Changed to: "Above the tropopause, a ~2 km thick temperature inversion is typically followed by a nearly isothermal temperature in the LS."*

39 'sharp distributions' - 'sharp gradients'

*Corrected!*

48 'an accurate representation of ... sharp gradients is of high importance for NWP' - a bit too strong 'may be of high importance' (other models with coarser vertical resolution still perform well)

*Corrected!*

61 'underestimated UT wind maxima ... in the ERA-15 reanalysis' - ERA-15 is quite old now, and coarse resolution by current standards, I think this should be mentioned. I think that the discrepancy has reduced (but not disappeared) in more recent versions.

*Thanks for this important remark! In our view, this sentence is also obsolete as it does not contain any new information about the representation of wind speed and wind shear. We have therefore removed this sentence.*

64 'satellite observations ... that DA smears out' - 'that satellite DA smears out'

It is well known that satellite soundings have limited vertical resolution (broad weighting functions, especially broad for microwave) and they are very numerous so this is not a surprise.

*We assume you are referring to the lines 73-74 on page 3. We revised the corresponding sentence as suggested!*

75-76 'Hence, no definitive conclusion can be drawn as to whether DA sharpens or smooths the tropopause.' My guess would be that it smooths slightly overall because the numbers of satellite soundings are so large. Also, if DA of RO and radiosonde data does sharpen the tropopause it raises the question of whether the DA has added detail that is inconsistent with the model dynamics at the current resolution. I suggest that the sentence be rewritten.

*We revised the conclusion from the indicated studies and rewrote the misleading sentence (p.3, ll. 73-75):*

*"Both studies, which show different effects of DA on the tropopause, differ in terms of the applied methods to diagnose the influence, the used observation type, the spatial resolution and the DA schemes."*

97 '[extra radiosondes] ... launched and applied in an OSE' replace 'applied', perhaps with 'used in an OSE' but 'their impact was studied in an OSE' would be more precise.

We revised the sentence as suggested.

110-113 'about 9200 radiosonde profiles'

Are these all high-resolution reports? A small proportion of reports in this area are only available at lower resolution (as for alphanumeric reports). The ship BUFR reports are at lower resolution than most of the land stations, but after the ECMWF vertical thinning there probably isn't much difference.

That is a good and important point, that we forgot to mention. 65 % of the profiles provide a lower resolved grid (< 100 levels, 300-400 m resolution) and 35 % of the profiles provide about 200-400 vertical levels or 100 m resolution (after thinning). We think that the revised description in Sec. 2.1 in combination with the supplementary Fig. S1 (see also discussion general comment #2) should makes this now clearer (see revised version pp.5-6, ll. 149-155):

*"It has to be noted that the radiosonde profiles are not assimilated at their fully measured vertical resolution (which would be ~5 m) but at a reduced number of levels (~50–350), which depends on the reporting type (e.g. alphanumeric, BUFR) the individual stations used for the data transmission to the Global Telecommunications System (GTS) (Ingleby et al., 2016). About 65 % of the assimilated radiosonde profiles used in this study have a low vertical resolution (<100 data points per profile) while 35 % of the profiles exhibit up to roughly 400 levels (see Fig. S1a in Supplement). Accordingly, the distribution of the average vertical distance of neighboring data point in the UTLS show a bi-modal shape and varies between ~100 and 400 m in the UTLS (Fig. S1b in the supplement)"*

115 'on demand' to 'on-demand'

Corrected.

119 'aircrafts' to 'aircraft' (yes, 'aircraft' is its own plural!)

Corrected! In line 121, the same error has also been corrected.

130 'With the aim' delete? Start with 'To investigate.'

Corrected.

137 'atmospherics state' - 'atmospheric state'

Corrected.

147 'using pressure used' - delete 'used'

Corrected.

216 'provides a high data coverage' - 'has a good data coverage'

Corrected.

246 'temperature decreases' - should be 'increases'?

Thank you for reading carefully. Of course, the temperature profile shows an increase in the lowermost LS. Corrected!

247 'above tropopause' - 'above the tropopause'

Corrected.

260 'a wind speed increase in the analysis' - this is what I would expect, see general comments.

Please see the discussion related to the general comments above.

Figure 6. The legend, especially the subscripts, is too small to read.

We increased the legend size, so it should be readable now.

308 'an uni-modal' - 'a uni-modal' (sounds right to me)

Corrected.

Figure 8. I am not sure that this figure adds much. To me a more interesting question (general comments) is the link, if any, between the tropopause sharpness and the maximum wind.

We consider Figure 8 important as it shows innovations and residuals that are indicators for the magnitude of background and analysis errors in the IFS. This information is needed for one of the key results of our paper, that DA sharpens the gradients across the tropopause, but the effect is small compared to model biases. Hence, we decided to keep Fig. 8. The link between tropopause sharpness and the maximum wind is evaluated and figures are added to the supplement (see comment above).

370 'changed the interval' delete 'the'

Corrected.

370 'exhibits' - 'exhibit'

Corrected.

383-384 'For the interval of smallest innovations, ... deteriorated tropopause altitude.' This is just a sampling effect. If you have an innovation of ~0, then the only way it can change in the analysis is to get larger in magnitude.

Thanks, we agree! We decided to remove this sentence, because this aspect was already mentioned three sentences earlier (page 16, lines 379-380).

416 'sharpen' - 'sharpens'

Corrected!

438 'other observations to influence' - 'other observations also influence'

Corrected!

440 'e.g., GPS radio occultation or dropsonde observations' I suggest 'aircraft or GPS radio occultation observations'. Dropsondes are too sparse (and sometimes dropped too low) to have much effect.

Thanks for your suggestion. We want to emphasize, that a larger number of dropsondes were released during NAWDEX over the northern Atlantic basin (Schäfler et al., 2018; Schindler et al., 2020) – that's why we mentioned the potential influence dropsondes. Nevertheless, we have included the aircraft measurements. The two sentences now read as follows (p.19, ll.451-453):

*"This may be due either to the remote impact of operational radiosondes or to dropsonde observations of which a larger number were deployed during NAWDEX (see Schindler et al., 2020). Further contributions of assimilated aircraft observations and GPS radio occultation data are also conceivable."*

474 'The remaining LS cold bias in the analysis (0.2 K) corresponds to previous assessments ...' Add that the main cause appears to be excessive humidity in the analyses at those levels - giving radiative cooling. This is mentioned in two of the references (Shepherd et al and Bland et al). Perhaps mention recent changes at ECMWF that have reduced, but not eliminated the cold bias: https://www.ecmwf.int/sites/default/files/elibrary/2021/19875-stratospheric-modelling-and-assimilation.pdf

Thank you for this comment and providing the reference Politchtchouk et al. (2021) and Ingleby et al. (2017). To incorporate your subsequent comment, we revised the whole corresponding paragraph (p.21, ll.487-494):

*"The remaining LS cold bias in the analysis (0.2 K) corresponds to previous assessments (Radnóti et al., 2010) and is driven by radiative cooling due to water vapor (Sheperd et al., 2018; Bland et al., 2021), which is systematically overestimated at those levels (Krüger et al., 2022). Recent changes at the ECMWF reduced but not fully removed the bias in the IFS (Polichtchouk et al., 2021). The warm bias (1 K) at the tropopause in the IFS was related to the finite vertical resolution of the IFS incapable of fully resolving the tropopause (Ingleby et al., 2016), the assimilation of warm-biased aircraft data at tropopause flight levels (Ingleby et al., 2017) and the moist bias in the LS of the IFS (Bland et al., 2021). The magnitude of the warm bias (about 1.2 K) at the tropopause is about 2-3 times stronger than the corresponding warm bias reported in Bland et al. (2021)."*

475-476 'The warm bias at the tropopause (1.2 K) is in line with Ingleby et al (2016). ... compared to Bland et al (2021).' - Needs rewriting. I'm not sure where the 1.2 K comes from. Ingleby et al (2016) has a statement "direct use of the tropopause significant level may result in a local bias (observation cooler than background)" but doesn't give a value for the bias.

I am also confused by the comparison to Bland et al which seems to say that they found a significantly larger bias. One factor is aircraft temperature bias and the many aircraft reports at 200 and 250 hPa. The following is from p 10 of Ingleby (2017): "At 200 hPa the O-B difference is more negative than at adjacent levels - this is due largely to a warm bias in flight level aircraft temperatures feeding through to the background fields. Figure 3.2 shows that at 200 hPa the background values at radiosonde locations are about 0.2° higher without aircraft assimilation"

Ingleby B. 2017: An assessment of different radiosonde types 2015/2016. ECMWF Tech. Memo. 807 (ECMWF website)

See previous comment.

484 'positive shear in the up to' - 'positive shear below'

We revised the whole paragraph to better describe increments of wind speed and wind shear. As a result, this sentence was removed.

502-504 'In case the LRT latitude of background and observation is comparable ... resolution of the IFS.' As mentioned above (see 383-384) this is a sampling issue and I recommend that it is deleted from this section.

We follow the reviewer's suggestion and deleted the sentence!

521 'routinely radiosondes' - 'routine radiosonde or aircraft data'

Corrected.

521 'at a close-by location' - 'nearby' (one word will do)

Corrected.

524-525 'the B-matrix ... spreads information ... horizontally and vertically'

This is true, but the vertical spreading is less important when assimilating a high-resolution profile from a radiosonde.

We agree that the horizontal spreading via B is certainly the more relevant factor here and removed "and vertically" from this sentence.

530 'strongly rely' delete 'strongly' (similar to comment on line 48).

Corrected.

542 'These increments are corresponding to' - 'These increments correspond to'

Corrected.

548-549 'sharper ... tropopauses, that are typically associated with ridge situations (high tropopause)'

Any background/evidence for this (I don't think it was mentioned earlier in the text).

For clarification of this issue we derived the average observed tropopause altitude for all classes of tropopause sharpness and added this information to Sect. 3.2.3 (p.14, ll.327-329 in the revised version):

*"In agreement with these findings the observed mean tropopause altitude for the sharp and smooth classes are 12750 m and 11580 m, respectively, which suggests that the sharp (smooth) tropopauses can be related to ridge (trough) situations characterized by high (low) tropopause altitudes."*

580-581 'The feedback files analysed ...'

These are not stored in MARS, I think someone (Gabor?) must have supplied them directly. The raw BUFR data (without feedback) are available from https://www.ncei.noaa.gov/data/ecmwf-global-upper-air-bufr/archive/

As the model space fields of the experiments were stored in MARS, the feedback files were also archived (See also: MARS catalogue, for example for the experiment ID gmgc): https://apps.ecmwf.int/mars-catalogue/?stream=oper&expver=gmgc&month=sep&year=2016&type=mfb&class=rd.

670-671 'Lavers ... Accepted' - now published online

Reference "Lavers et al. 2023" has been updated.

---

## Author Comment (AC2)

MS No.: egusphere-2023-2094 – Influence of radiosonde observations on the sharpness and altitude of the midlatitude tropopause in the ECMWF IFS

By Krüger et al. (2023)

Reply to review #2

Dear Reviewer, we are grateful for your positive review of our manuscript, the appreciation of our study and that you recommend it for publication in WCD. Your comments helped us to improve the manuscript. Below, we answer each particular comment using a blue font. We also added a revised version that includes all corrections using track changes.

**General comments**

The paper addresses the representation of tropopause sharpness in forecast data and studies the effect of radiosonde observations on the assimilated tropopause structure.

For this purpose, the authors use more than 9700 radiosonde profiles in autumn 2016. Out of these 500 sondes were released as additional soundings in the frame of the NAWDEX experiment. These are used for an IFS observing system experiment with and without these additional soundings. For the full data set the authors analyze the emerging increments, innovations and residuals of temperature, wind (as well as shear) and static stability. Importantly they do this in tropopause relative coordinates to extract the effect of the assimilation of additional soundings on the tropopause thermal structure and winds.

In general their analysis clearly shows that the sondes lead to a sharpening of the tropopause in the assimilation. They further split the data according to Brunt Väisälä frequency in sharp, smooth and medium gradient tropopause cases and show that the sharpest tropopauses require the strongest increments, similar for the winds.

Overall, they found a sharpening of the tropopause with increased $N^2_{max}$ and increased shear values from positive at the wind maximum to negative above the tropopause. In particular they infer from a comparison with and without the additional sondes a substantial contribution of the additional sondes to the assimilation. They also show that the analysis tropopause altitude is shifted towards the sounding observations. The comparison of the OSE runs highlights that the main contribution to the tropopause sharpening can be attributed to the radiosondes.

The only point which could be discussed by the authors is the role of humidity as possible reason for the temperature deviations at the tropopause (see below), though the humidity is not assimilated, it might explain at least partly the discrepancies of tropopause sharpness compared to the observations.

Overall the paper is very clear, well-structured and each analysis step is clearly motivated. The methods are well documented and appropriate, the emerging conclusions are scientifically sound - it was a pleasure to read.

The paper clearly merits publication in WCD and I see only minor points.

l.100: Although moisture is not assimilated the incorrect representation in the IFS, it may lead to larger temperature differences above e.g. cirrus clouds compared to clear sky observations. Cirrus occurrence in observational data might be misrepresented or missing in the IFS data, particularly for the $N^2\_max$ cases (i.e. ridge regions). Humidity is not assimilated and therefore not analyzed by the authors. Nonetheless it could be discussed (maybe in the final discussion) as possible cause for the misrepresentation of temperature at the tropopause. Would it be possible to relate the temperature increment at the sounding location to the observed humidity compared to the background humidity? A larger increment for different saturation conditions for IFS versus sounding would provide a potential explanation of temperature increments at higher tropopauses.

We agree with the reviewer that it would be very interesting to further study the connection between temperature and humidity errors in the UTLS. In principle, the passive (= not assimilated, but contained in the files) humidity data is also monitored, which we did not consider in this study. Using such data would allow to correlate temperature bias and increments with the cloud and moisture at the tropopause. Such an investigation would be feasible; however, it is beyond the scope of this study (temperature and wind influence of data assimilation). In our manuscript we refer to the study by Bland et al. (2021) at several points which provides a detailed analysis about the relation of temperature and moisture errors at and above the tropopause. Please note that we revised the discussion following a comment by the other reviewer, which we hope also addresses your comment (p.21, ll.487-494):

*"The remaining LS cold bias in the analysis (0.2 K) corresponds to previous assessments (Radnóti et al., 2010) and is driven by radiative cooling due to water vapor (Sheperd et al., 2018; Bland et al., 2021), which is systematically overestimated at those levels (Krüger et al., 2022). Recent changes at the ECMWF reduced but not fully removed the bias in the IFS (Polichtchouk et al., 2021). The warm bias (1 K) at the tropopause in the IFS was related to the finite vertical resolution of the IFS incapable of fully resolving the tropopause (Ingleby et al., 2016), the assimilation of warm-biased aircraft data at tropopause flight levels (Ingleby et al., 2017) and the moist bias in the LS of the IFS (Bland et al., 2021). The magnitude of the warm bias (about 1.2 K) at the tropopause is about 2-3 times stronger than the corresponding warm bias reported in Bland et al. (2021)."*

Fig.2 and related discussion: How does the altitude distribution of the 500 additional sondes compare to the rest? Could you add the PDF for those additional 500 soundings as separate contour?

That's a valid point. In the revised version of the manuscript, we added the tropopause altitudes distribution of the additional NAWDEX sounding (see the following figure). It clearly shows that due to the lower number of profiles at latitudes < 40°N less high tropopause altitudes (14-15 km) was observed.

[Figure]

**Figure 2:** Stacked distribution of LRT$_{yO}$ with 0.2 km bin size for (a) all 9729 radiosondes and (b) the additional 497 radiosondes observed during NAWDEX. The colouring shows the latitude of the radiosonde stations (10° bins).

In the revised manuscript we included the following paragraph to describe that (p.7, ll. 200-206):

*"The left mode represents profiles with a high frequency (75 % of the profiles) of LRT altitudes at 10-14 km (see Fig. 2a) which is typical for the midlatitudes in autumn (e.g., Hoffmann and Spang, 2022; Krüger et al., 2022). Its broad spectrum is related to the variability of the midlatitude tropopause in different synoptic situations, e.g. in ridges and troughs (Hoerling et al., 1991). The right mode (LRT > 14 km; 25 % of the profiles) with its smaller maximum indicates profiles in the subtropics. The LRT distribution for the additional NAWDEX radiosondes (Fig. 2b) does not exhibit a corresponding second peak, due to the low number of soundings conducted at latitudes < 40 °N."*

*Hoerling, M. P., Schaack, T. K., and Lenzen, A. J.: Global Objective Tropopause Analysis, Mon. Weather Rev., 119, 1816–1831, https://doi.org/10.1175/1520-0493(1991)119<1816:GOTA>2.0.CO;2, 1991.*

Since the subtropical tropopause and the extratropical tropopause have partly different drivers, how do the results change when only considering extratropical tropopauses with altitudes less than 14000 m? Wouldn't one expect different effects of the assimilation of the mainly extratropical 500 soundings for the extratropical tropopause compared to the subtropical tropopause?

Thank you for this comment. A separate analysis for the sub-tropics and mid-latitude was also suggested by the other reviewer: Based on the observed tropopause altitudes in Fig. 2 we define profiles with an LRT >14 km as sub-tropical (25 % of the profiles) and LRT <14 km as mid-latitude profiles (75 %). The tropopause-relative profiles of observed temperature, N², wind and wind shear as well as the increments are shown for both classes in Figure S2. Compared to the mean midlatitude profiles which shows similar distributions as for the overall data set (compare Fig. 5). The sub-tropical profiles exhibit a considerably lower temperature in the entire UTLS, a weaker LS temperature inversion, a cooler tropopause and furthermore show continuously decreasing wind speed with altitude and no wind maximum being located near the tropopause. This represents a typical temperature and wind distributions one might expect poleward of the sub-tropical jet. We consider this an interesting finding relevant for the reader and decided to add Fig. S2 to the Supplement. In addition, we added the following description in Sect. 2.2 (p.9, ll.226-228 in the revised version):

*"A separate analysis of extratropical (LRT < 14 km) and sub-tropical (LRT > 14 km) observations reveals similar shapes for the extratropical and the overall data (see Fig. S2a-d). The sub-tropical mean profiles*

*exhibit lower temperatures in the entire UTLS, a weaker temperature inversion in the LS and no wind maximum being located near the tropopause.”*

[Figure]

Figure S2: LRT$_{yO}$–relative mean profiles of (a) temperature, (b) N², (c) wind speed, (d) wind shear for profiles associated with the mid-latitudes (LRT$_{yO}$ < 14 km; solid) sub-tropics (LRT$_{yO}$ > 14 km; dashed).

In addition, we provide a plot for the average increments (Fig. S3). Increments of sub-tropical profiles are weaker, but still point in the same direction as in the midlatitudes. The wind speed increments are smaller in the upper troposphere at lower wind speeds. We added Fig. S3 to the Supplement and a description to Sect 3.2 (p.11, ll. 270-272 in the revised version).

*“A separate analysis of mid-latitude and sub-tropical increments (Fig. S3) shows that the latter are weaker. However, as the increments in both regions point in the same direction, the complete data is considered for the statistical analysis in the remainder of this article.”*

[Figure]

Figure S3: as in Fig. S2 but for increments.

l.43: Please also refer to the work of Kaluza et al. 2021 (WCD) who showed the existence of a shear layer in tropopause relative coordinates in ERA5.

     References: Kaluza, T., Kunkel, D., and Hoor, P.: On the occurrence of strong vertical wind shear in the tropopause region: a 10-year ERA5 northern hemispheric study, Weather Clim. Dynam., 2, 631–651.

We are grateful for providing this reference. In l.43 (p1) we describe the tropopause structure as observed by a radiosonde climatology according to Birner et al. (2002). However, we included this reference in the Discussion (p.22, ll. 505-509):

*“The observed vertical wind shear profile is characterized by positive values below and negative above the wind maximum as well as by a sharp increase of negative shear across the tropopause. The enhanced (negative) shear in the 1 km layer above the tropopause in the observations is also present*

*in the ECMWF, which is consistent with previous findings (Schäfler et al., 2020; Kaluza et al., 2021); its magnitude, however, is considerably weaker in the background and analysis as compared to the observations."*

l.484/485: Sentence reads strange, please rephrase.

We revised the whole paragraph to better describe increments of wind speed and wind shear. As a result, this sentence was removed.